# Reconciling crop production, climate action and nature conservation in Europe by agricultural intensification and extensification

Ting Hua [1] ✉, Xiangping Hu [1], Gunnar Austrheim [2], James D. M. Speed [2], Bob van Oort [3] & Francesco Cherubini [1]

Agricultural production in areas characterized by low productivity, steep slopes, and high fragmentation is usually associated with higher-than-average management costs and environmental impacts. Abandoning this suboptimal cropland to vegetation regrowth, while optimizing crop production in other locations, is an attractive strategy for supporting climate and biodiversity targets without compromising food security. However, it has not yet been explored within the specific context of European agriculture. Here, we identify the area extent of suboptimal cropland in Europe and assess if crop production losses from its revegetation can be compensated by implementing scenarios of cropland intensification or extensification elsewhere. We found 24.2 million hectares of suboptimal cropland, of which 66% is at degradation risk and about 50% is within biodiversity priority areas. Reducing agricultural intensity in 16.4–30.9 million hectares of the remaining cropland by introducing parcels of trees into the agricultural landscape (extensification), together with strategic crop-switching optimization, can entirely offset crop production losses from revegetation of suboptimal cropland. This scenario has the potential to mitigate up to 40% of European agricultural emissions of greenhouse gases and reduce cropland pressure on biodiversity by 20%. In contrast, cropland intensification achieves lower carbon-biodiversity benefits, with risks that crop losses are not fully compensated.

Agriculture and food systems account for roughly one-third of global anthropogenic greenhouse gas (GHG) emissions[1], are the main driver of biodiversity loss[2] and freshwater withdrawals[3], and induce various forms of environmental pollution (e.g., from excessive use of agrochemicals)[4]. Simultaneously, agricultural production and the resilience of the food system are sensitive to environmental degradation[5] and climate change[4], and their functioning highly depends on natural resources and processes. A system transformation of the agricultural sector is therefore necessary to reconcile crop production with climate change mitigation, adaptation, and nature conservation[6–8].

Sustainable intensification, extensification, switching crop type and relocating agricultural areas are among the key strategies proposed to secure sustainable yields while sparing land for natural

---

[1]Industrial Ecology Programme, Department for Energy and Process Engineering, Norwegian University of Science and Technology, Trondheim, Norway. [2]Department of Natural History, NTNU University Museum, Norwegian University of Science and Technology, Trondheim, Norway. [3]CICERO Center for International Climate Research, Oslo, Norway. ✉e-mail: ting.hua@ntnu.no

habitats and carbon sequestration[9–12]. Each option has benefits and adverse effects that primarily depend on local context and management practice, and may encounter various technical, cultural and socio-economic barriers for implementation[13,14]. In general, agricultural intensification aims to increase crop production in situ, often resulting in the abandonment of cropland in areas with low productivity or high conservation value, thereby enabling the expansion of natural vegetation[15,16]. Evidence suggests that cropland intensification has the potential to increase global crop production by 45 to 70%[17–19], or to maintain current crop production volumes using half of present-day cropland areas through optimizing cultivation practices[7,11]. However, there are risks of accelerating soil degradation, environmental pollution, and biodiversity loss due to high-density cultivation and intensive use of agrochemicals[20–22]. Agricultural extensification, a concept close to landscape diversification and multifunctionality, aims to decrease the environmental impacts of agricultural activities and reduce exploitation of natural resources by improving the conditions that support habitat diversity and ecosystem services within agricultural landscapes, while minimizing possible declines in crop production[23,24]. For example, integration of trees into cropland can promote multifunctional landscapes with cascading benefits on yield through increased soil nutrient and water retention, reduced soil erosion, and improved climate resilience[25–27]. In such cases, crop switching to high-yield varieties is typically needed to compensate for the reduction in available cultivation areas, and to maintain total crop supplies[28].

In Europe, agriculture covers 38% of the total land area[29]. Cereals (mainly wheat, maize and barley) occupy roughly one-third of the agricultural land, accounting for a quarter of its crop production value[30]. The agri-food sector also contributes 31% of total GHG emissions[31], and it is the most frequently reported threat to both habitats and species[32]. There are policies and initiatives that support the transition to more sustainable farming systems that mostly aim at reducing agrochemical inputs and improving biodiversity, such as the European Green Deal (EGD)[33], the European Union (EU) biodiversity strategy for 2030[34] and the Common Agricultural Policy (CAP)[35]. Implementing these policies can alleviate the environmental pressure from agriculture in Europe, but risks to reduce crop production within Europe and simultaneously induce ecosystem impacts elsewhere by increasing crop imports. Achieving the agricultural target for 2030 in the EGD is projected to reduce European food self-sufficiency and increase the demand for agricultural land outside the EU of about 24 million hectares (Mha), resulting in land-use-related emissions of 759 million tonnes (Mt) $CO_2$-equivalent ($CO_2$-eq.) and a biodiversity impact of 3.86 million mean species abundance loss[36]. Identifying cropland management strategies that can deliver carbon-biodiversity benefits without compromising crop supply in Europe can prevent these risks.

Agricultural production in suboptimal cropland, i.e., areas with low productivity and/or reduced economic return because of severe local constraints on agricultural use, typically requires targeted subsidies and intensive management to attain competitive yields, and it is frequently associated with environmental impacts, land degradation and high cultivation costs[37,38]. These issues are expected to further intensify under future climate change[39]. Promoting expansion of conservation and restoration measures on suboptimal cropland[40,41], together with implementation of in-situ solutions in other cropland landscapes to secure stable yield levels[42–44], is one possible strategy to reconcile agriculture with climate action and conservation goals. Although evidence of the environmental benefits of these individual strategies is consolidating[45–47], our understanding of their integration potential and the biophysical implications for crop production, climate change mitigation, and biodiversity conservation remains limited, preventing the identification of specific land-transformation targets for Europe.

Here, we introduce a continental-scale approach to identify potential suboptimal cropland in Europe, integrating criteria of low productivity, high fragmentation and steepness of the terrain, and investigate various solutions to compensate for the decrease in crop production from its revegetation. We consider eight scenarios that combine revegetation options of suboptimal cropland via passive natural regrowth[48] or active afforestation[49], adjustments in cropland use modes on the remaining cropland (via intensification or extensification), and crop switching strategies (Fig. 1). For all scenarios, we perform a bottom-up quantification of the effects on total crop production, area occupied by cropland or trees, and evaluate benefits and trade-offs for climate change mitigation and biodiversity conservation. In agricultural intensification, low-density cropland (i.e., landscapes with mixed parcels of crops and trees or shrubs) is converted to high-density cropland (i.e., landscapes made of pure cropland), while the opposite occurs in the extensification scenarios. Adjustments in crop type distributions consider 11 major crop types, commonly cultivated across Europe and suitable for respective agro-climatic zones. An agro-ecological model[50] that accounts for multiple local biophysical constraints and climatic conditions at a 5 arc-min resolution is used to estimate crop yields under different crop switching strategies, which involve changing the current crop to either the one with the highest calorie supply or, to facilitate farmer's acceptance and match local food preferences, the crop with the highest local suitability (intended as the crop with the today's largest harvest area in each pixel). Various thresholds for land-use constraints are considered, such as excluding cropland intensification in areas of biodiversity priority[51] and at risks of water scarcity[52] or land degradation[38] (identified by the presence of different types of degradation processes that impact agricultural productivity), wherein cropland extensification is prioritized. Climate change mitigation benefits account for carbon sequestration by trees in the revegetated suboptimal cropland and in the extensification scenarios, carbon emissions from land clearing in the intensification scenarios and from afforestation activities. Climate impacts also consider direct and indirect changes in soil $N_2O$ emissions[53] caused by changes in fertilize use after abandonment of suboptimal cropland and crop switching in the intensification/extensification scenarios. Biodiversity impacts are estimated by comparing local species richness under natural vegetation (or planted forest) versus cropland[54,55]. These indicators are finally integrated within a land-climate-biodiversity nexus to illustrate co-benefits and trade-offs of the investigated scenarios.

## Results
### Suboptimal cropland

Cropland located on steep slopes (> 8°), fragmented or isolated (individual cropland patch < 10 ha) and with relatively low agricultural productivity is widespread across Europe, totaling 24.2 Mha, or about 13.7% of total European cropland (Fig. 2, see "Methods" and Supplementary Text 1 for more details on definition and identification of suboptimal cropland). This value includes both pure cropland pixels (20.0 Mha) and cropland fraction in mosaic cropland pixels (4.20 Mha), where about 50% of the land is cropland. Supplementary Figs. 1, 2 depict the fraction of suboptimal cropland relative to the total cropland in each pixel and to the national total cropland area, respectively. High concentrations of suboptimal cropland are observed in southern and eastern Europe, with 4.65 Mha in Italy, 3.36 Mha in Spain, and multiple clusters across the Balkans.

Most of this suboptimal cropland (19.7 Mha, or 81.4% of the total) is in steep slope terrains (Fig. 2b), while the fragmented cropland is 0.70 Mha, mainly concentrated in central Europe. About 4 Mha of low-productive cropland, identified using an approach that measures vegetation productivity during the growing season with remote sensing data, is scattered throughout Europe. In Norway, Italy, Greece and

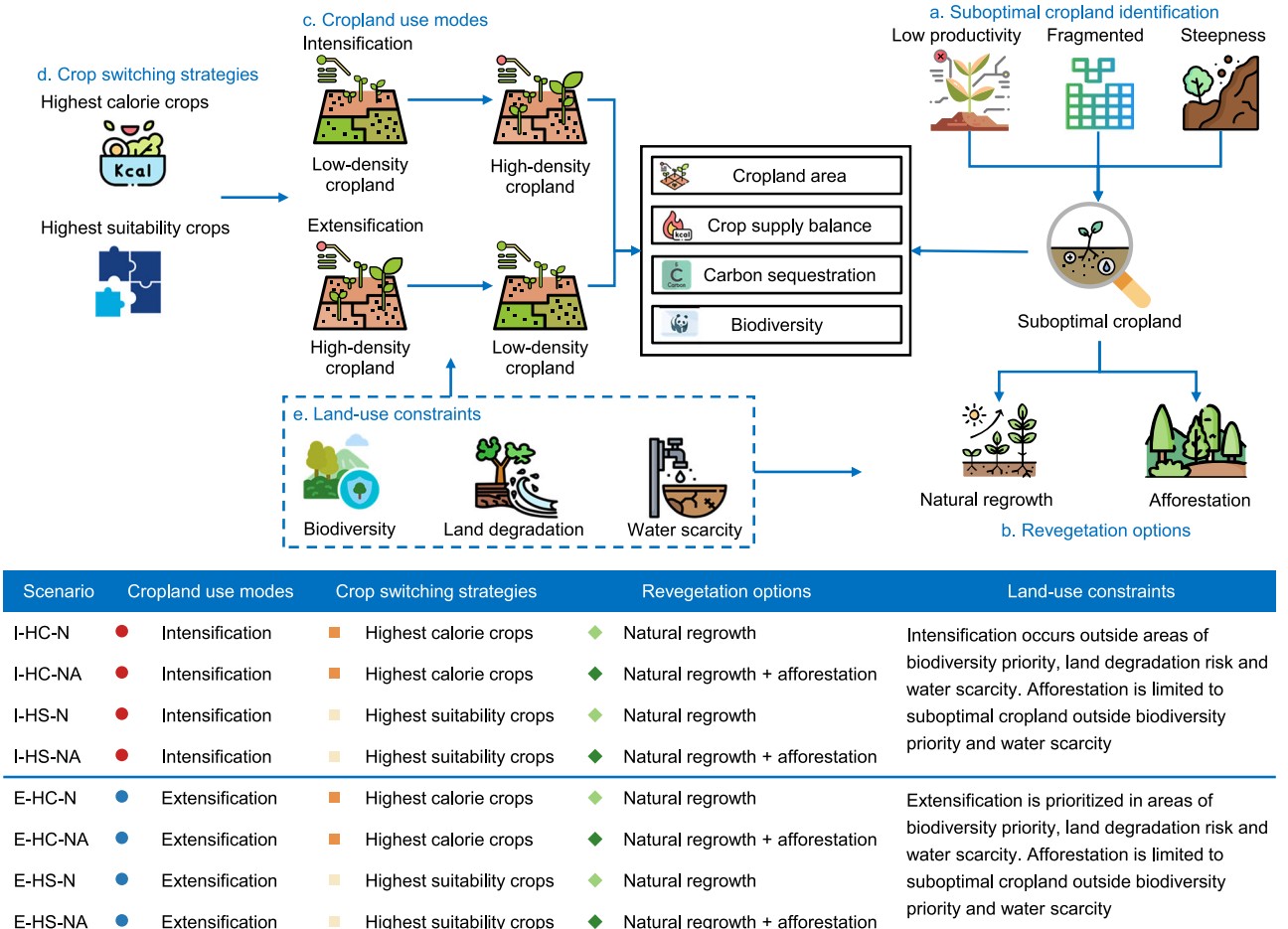

**Fig. 1 | Schematic representation of the research framework used to estimate suboptimal cropland and assess cropland management strategies to advance the land-climate-biodiversity nexus in Europe.** Suboptimal cropland is identified by considering cropland that is currently characterized by low productivity, high fragmentation, or high steepness gradients (**a**). This suboptimal cropland is revegetated through natural vegetation regrowth or afforestation to support nature conservation strategies and favor carbon sequestration (**b**). This induces changes in the total amount of cropland area in Europe and a crop supply imbalance, together with benefits for climate change mitigation and biodiversity. The crop supply imbalance is complemented by adjusting cropland use modes (that is, via intensification or extensification of the remaining cropland) and crop switching, either considering the crop with the highest calorie supply or the one with the highest suitability (defined as the crop type that is more common in a given area) (**c**, **d**). Various land-use constraints are considered for cropland use mode conversion and revegetation options (**e**). The bottom table shows the eight investigated scenarios that are a combination of cropland use modes (I: intensification; E: extensification), crop switching strategies (HC: crops with the highest calorie supply; HS: crops with the highest suitability), and revegetation options (N: natural regrowth; NA: combination of natural regrowth and afforestation). Icons are sourced from Flaticon.com.

some countries in the former Yugoslavia, suboptimal cropland represents more than 20% of the national cropland area (Supplementary Fig. 2), and it is primarily due to cropland in steeply sloping terrain. In Norway, there is also a relatively high presence of fragmented cropland. In general, low-productive cropland constitutes between 1 to 10% of total cropland area across the European countries.

Despite representing nearly 14% of total European cropland, suboptimal cropland contributes with about 9.88% to the total dry-weight mass of crop production (9.85% in terms of calorie supply), according to the spatially explicit yield dataset Global Agro-Ecological Zones (GAEZ + 2015)[50] (Fig. 2e). The reduced average productivity can be attributed to the lower effectiveness of agriculture on steeply sloping terrain and terraces, where nutrient leaching is typically high, and risks of other land degradation processes are elevated[56,57]. Still, suboptimal cropland accounts for over 20% of the national calorie supply from investigated crops in countries with complex topography, such as Switzerland and several south-eastern European countries (Supplementary Fig. 3). The lower productivity of suboptimal cropland is confirmed when another spatial-explicit crop yield database is used[58], which shows an even lower average estimate (7.61% of the crop

dry-weight, Supplementary Fig. 4). The differences between the two databases primarily arise from variations in data sources, downscaling method, and mapping of irrigation[59,60].

About 69% of the identified suboptimal cropland overlaps with areas currently undergoing some form of degradation (Fig. 3a). The major threats are soil salinization, soil organic carbon (SOC) depletion, and soil acidification. Overall, 66% of the suboptimal cropland can be classified as being at risk of degradation, defined as land threatened by at least two co-occurring degradation processes[38], and about 53% is simultaneously affected by four or more degradation processes (Fig. 3b, c). About half of the suboptimal cropland also falls within the top 30% of global biodiversity priority areas (Fig. 3d), i.e., land that, if optimally managed for conserving biodiversity, can meet conservation targets for 81% of the considered terrestrial plant and vertebrate species[51]. A smaller fraction (10.1%) of suboptimal cropland is located in areas affected by blue water scarcity[52], indicating an insufficient availability of renewable freshwater resources for agriculture in surface and groundwater. Overall, abandoning agricultural production in suboptimal cropland has the potential to support climate action and conservation goals while mitigating land degradation, but

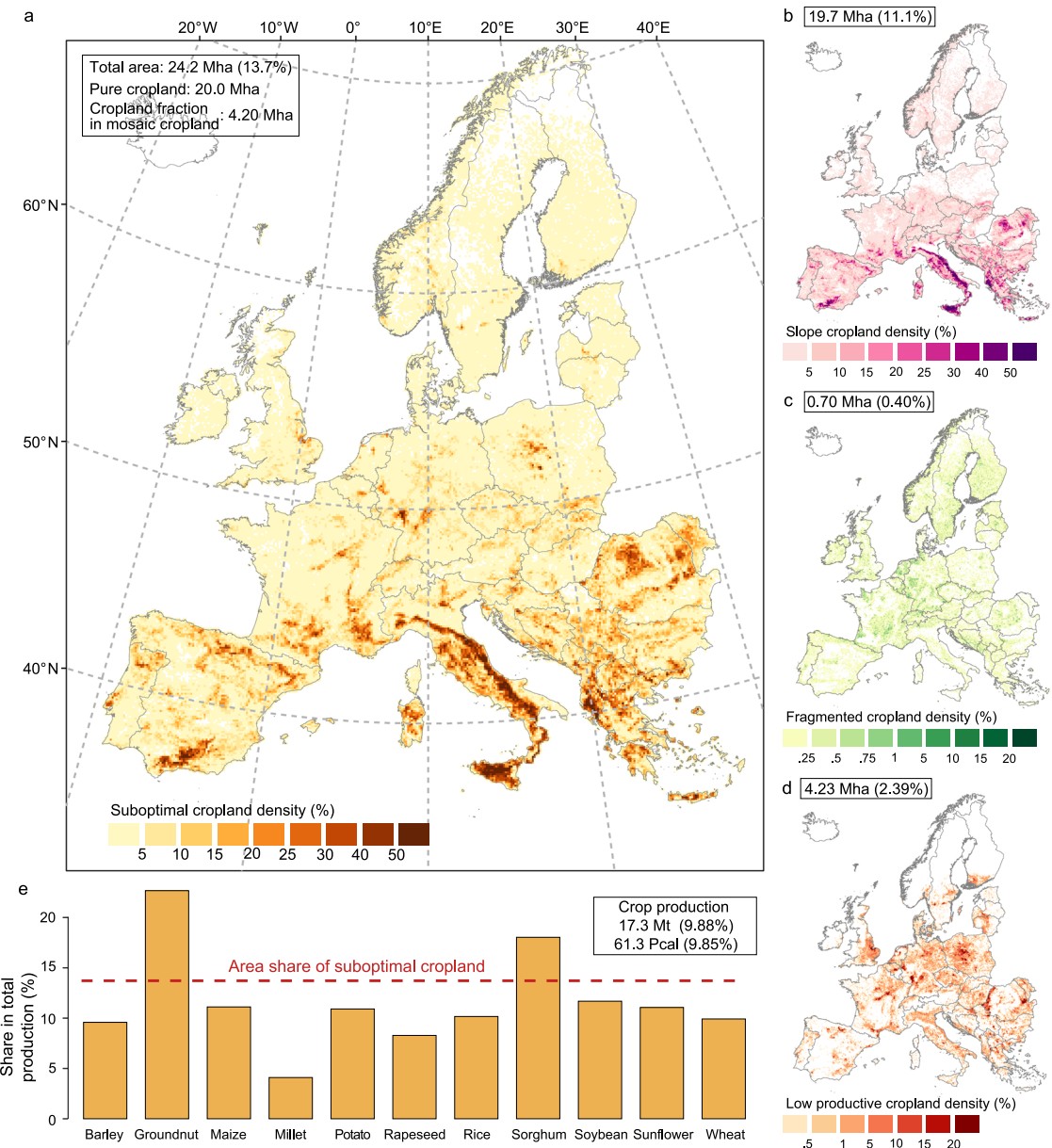

**Fig. 2 | Suboptimal cropland distribution in Europe and its relative contribution to total crop production.** The total identified suboptimal cropland (**a**) combines cropland in high slope terrains (**b**), fragmented cropland (**c**) and low productive cropland (**d**). In each of these subfigures, the numbers in the top left corner indicate the corresponding area and the share relative to the total cropland area in Europe. The total area of suboptimal cropland is 24.2 Mha (million hectares), comprising 20.0 Mha of pure cropland pixels and an additional 4.20 Mha contributed by the cropland fraction within mosaic cropland pixels. The color bar refers to the fraction of suboptimal cropland's area relative to the pixel area. Maps are aggregated to 10 km resolution for better visualization, from the original resolution of about 300 m. In (**e**), the shares of the production in suboptimal cropland of the major crops relative to total production in Europe are shown (Mt: million tonnes, aggregated on dry-weight basis; Pcal: Peta-calorie). The red dashed line indicates the area of suboptimal cropland relative to the total cropland (13.7%). When the orange bar is below this line, the productivity of the suboptimal cropland is lower than the average productivity of that specific crop at the European level. The total crop production considers the total dry mass of cereal crops, oil crops, sugar crops and root crops cultivated in suboptimal cropland, and it does not include the mass of vegetables and pulses. The estimation of total calorie supply is based on cereal crops, oil crops, sugar crops, root crops, vegetables and pulses. Basemaps in (a-d) are sourced from GADM, with Austrian data licensed under Creative Commons Attribution-ShareAlike 2.0 (source: Government of Austria).

complementary strategies should be explored to prevent possible adverse effects on food security at a continental level and spillover impacts at a global scale.

## Cropland intensification or extensification

Of the 24.2 Mha of suboptimal cropland, about 7 Mha land for woody perennial crops (e.g., olive and fruit trees) is retained, while the rest (17.1 Mha) is dedicated to revegetation via either natural vegetation regrowth alone, or by implementing natural vegetation regrowth within biodiversity priority areas and areas affected by water scarcity

(8.81 Mha), with afforestation elsewhere (8.30 Mha) (Supplementary Fig. 5). Such revegetation of suboptimal cropland leads to a crop production loss of about 61 peta-calories (Pcal), primarily comprising of 6.77 Mt dry-weight of wheat, 3.81 Mt of maize, and 2.51 Mt of barley. Alternative approaches combining two cropland use modes (intensification or extensification) with two crop switching strategies are explored to potentially offset this calorie loss (see Fig. 1 for the full list of scenarios and Supplementary Fig. 6 for the method design). Today's cultivation of olives, vegetables, nuts, fruits and pulses is maintained in both intensification and extensification

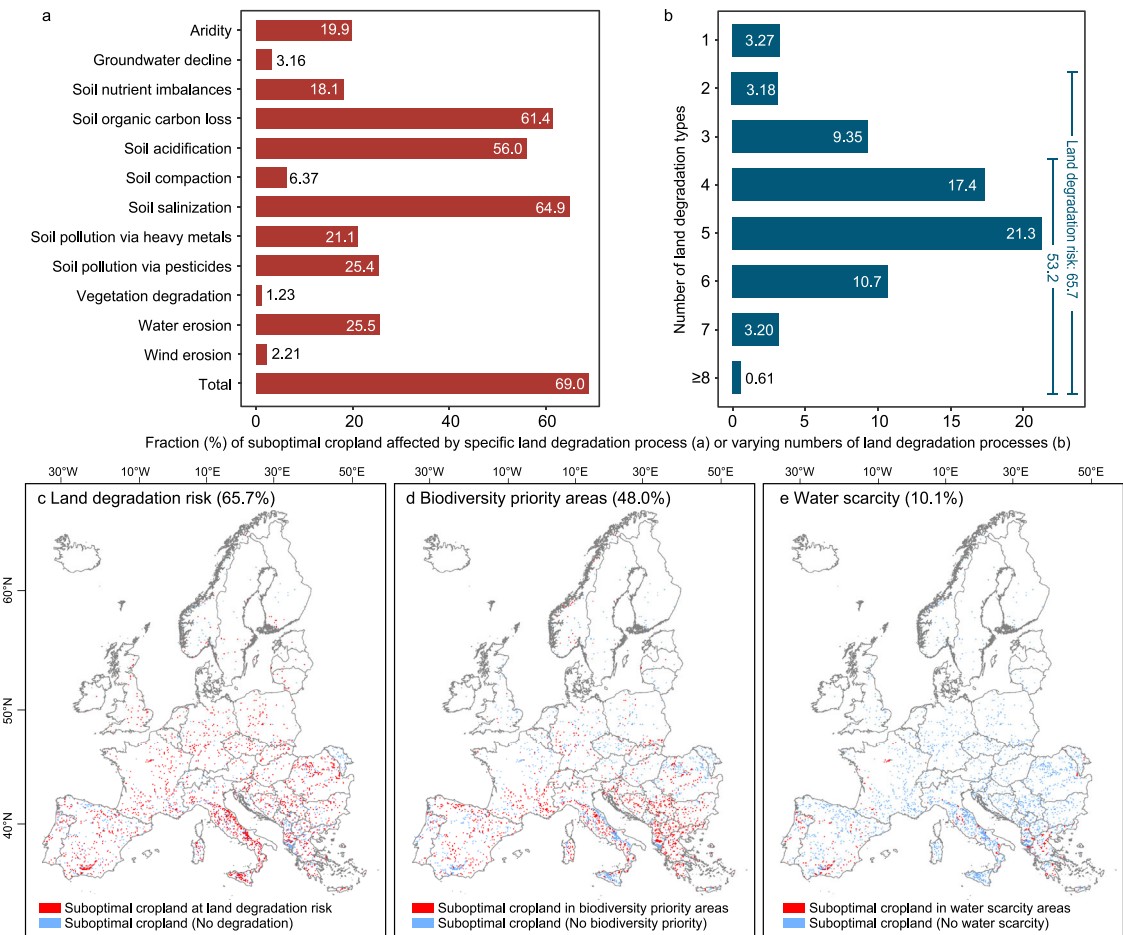

**Fig. 3 | Distribution of suboptimal cropland in areas affected by various land degradation processes, biodiversity priority areas, and areas threatened by water scarcity.** Fraction of suboptimal cropland in areas with different types of land degradation processes (**a**). 'Total' indicates the fraction of suboptimal cropland affected by at least one type of land degradation process. Fraction of suboptimal cropland affected by varying numbers of land degradation processes, relative to total suboptimal cropland (**b**). For example, 3.27% of suboptimal cropland is affected by only one land degradation process and 53.2% of suboptimal cropland is affected by at least four types of land degradation processes. 65.7% of suboptimal cropland is at risk of land degradation, defined as being threatened by at least two co-occurring land degradation processes[38]. Distribution of suboptimal cropland at land degradation risk (**c**), biodiversity priority areas (**d**) and areas affected by water scarcity (**e**). Maps are aggregated to a 10 km resolution for better visualization, from the original resolution of about 300 m at which the analysis is performed. Basemaps in (**c**–**e**) are sourced from GADM, with Austrian data licensed under Creative Commons Attribution-ShareAlike 2.0 (source: Government of Austria).

scenarios to retain the supply of key nutrients and existing vegetation carbon stocks.

In cropland intensification, low-density cropland (represented by mosaic cropland pixels that contain both cropland and non-crop vegetation) is converted into high-density cropland (i.e., pure cropland pixels). Intensification is prioritized for the mosaic cropland pixels that have a higher number of adjacent pure cropland pixels, until the calorie loss from revegetating suboptimal cropland is fully compensated. Areas of biodiversity priority[51], at steep elevation gradients, and at risk of land degradation[38] and water scarcity[52] are excluded from intensification. Crop production within the whole intensified cropland pixels is subject to switch to the crop with the highest calorie supply or local suitability, while the rest of the cropland remains unchanged.

The crop production losses from revegetating suboptimal cropland can be fully compensated by intensifying 4.78 Mha of mosaic cropland when cultivating crops with the highest calorie supply (I-HC in Fig. 4a). Most of the intensification occurs in the central band of Europe, where cropland is already intensively managed and biodiversity priority areas (a constraint to the intensification) are less concentrated than in southern Europe. This approach largely involves cultivation of maize (2.85 Mha), followed by wheat (0.95 Mha) and barley (0.29 Mha) (see Supplementary Figs. 7, 8 for the crop-specific cultivation area and distribution). Alternatively, switching to crops with the highest local suitability (i.e., the predominant types currently cultivated) and intensifying all available mosaic cropland (I-HS) could require 6.84 Mha and offset 91.1% of the calorie loss (Fig. 4b). This is due to constraints on available land for intensification from biodiversity conservation, land degradation and water scarcity. I-HS further expands intensification in central Europe and some Baltic countries, with cultivation of wheat becoming dominant (4.05 Mha), followed by barley (1.12 Mha) and maize (0.73 Mha).

In cropland extensification, high-density cropland (i.e., pure cropland pixels) is converted into low-density cropland (i.e., mosaic cropland), with priority given to areas that are important for biodiversity, affected by water scarcity and threatened by land degradation. Crop switching strategies are implemented in the cropland fraction of the new mosaic cropland. The analysis considers the increasing evidence of the generally positive effect of cropland extensification on crop yield (see Supplementary Table 1), as the presence of trees contributes to regulate the microclimate, mitigate weather extremes, improving pest control and retain soil water and

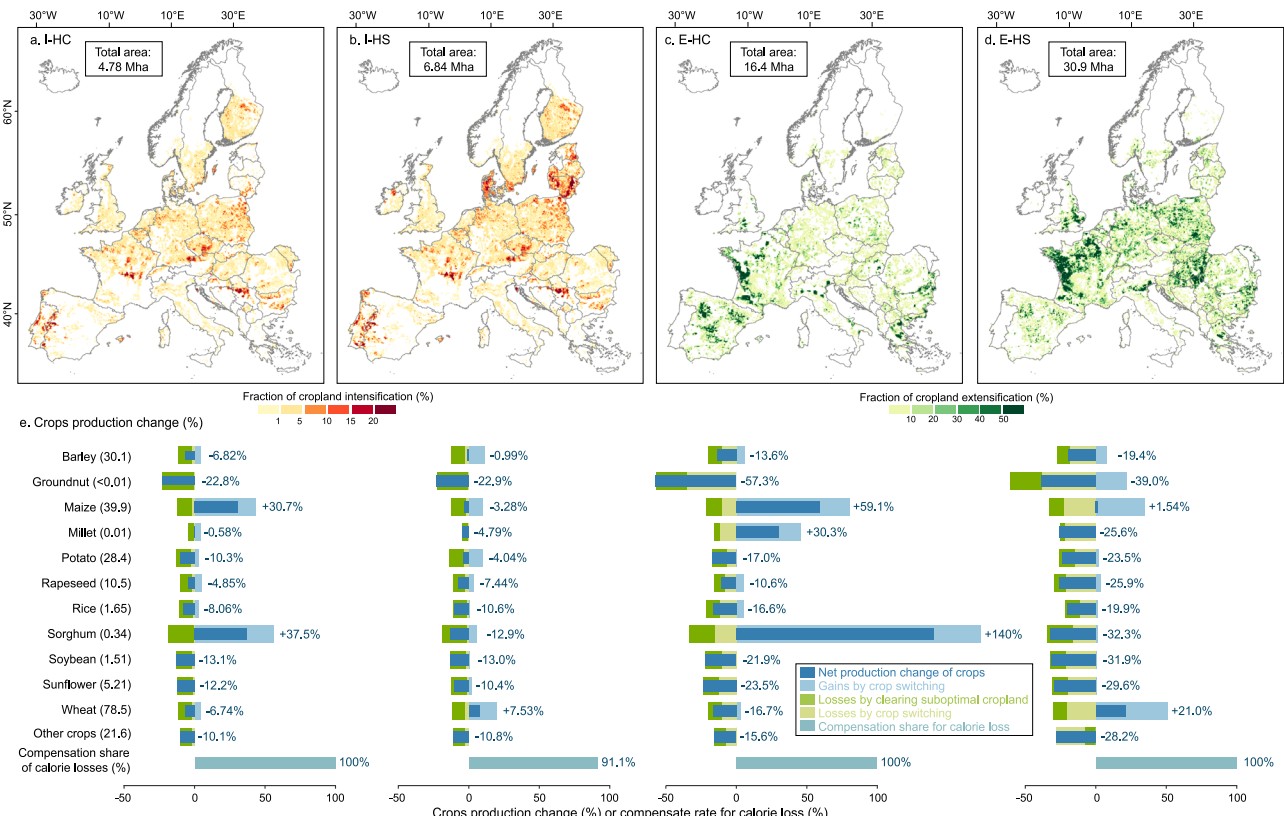

**Fig. 4 | Cropland intensification and extensification scenarios to balance crop production losses from revegetation of suboptimal cropland in Europe.** Distribution of cropland under four contrasting approaches combining cropland use mode (I: cropland intensification, E: cropland extensification) with crop switching strategies (HC: crops with the highest calories supply, HS: crops with the highest suitability, Mha: million hectares) (**a**–**d**). The color bar in these maps refers to the fraction of cropland intensification or extensification relative to the total pixel area. Values in the box indicate the land area that requires conversion for intensification or extensification. Maps are aggregated to 10 km resolution for better visualization, from an original resolution of about 300 m at which the analysis is performed. **e:** changes in crop production under the scenarios investigated. The numbers in parentheses indicate the present-day production for 11 major crops (fresh weight, in million tonnes) and other crops (i.e., sugar crops and other cereals, aggregated on a dry-weight basis) from all croplands in Europe. Numbers in blue inside the bar charts indicate the net production change after revegetating suboptimal cropland, cropland use mode conversion, and crop switching. The different colors of the stacked bar carts reflect net changes in crop production (dark blue), crop production gains (light blue) and losses (green) in crop switching, and production decreases from clearing production from suboptimal cropland (light green). The bottom bar shows the crop compensation share of the investigated scenarios, defined as the ratio of net calorie gains from crop switching relative to the calorie losses from revegetating suboptimal cropland. For example, I-HC, E-HC and E-HS fully compensate for the calorie losses, while I-HS can compensate for only 91.1%. Basemaps in (**a**–**d**) are sourced from GADM, with Austrian data licensed under Creative Commons Attribution-ShareAlike 2.0 (source: Government of Austria).

nutrients[23,24,61,62]. Yield effects on the cropland fractions within the new mosaic pixels were estimated using an unmixing method applied to a high-resolution yield dataset[63] across 164 main subregions in Europe (Supplementary Fig. 9 for a representation of the subregions), revealing a continental average yield benefit of about 9.5% (Supplementary Text 2 and Supplementary Fig. 10 for methodological details).

Crop production losses from revegetation of suboptimal cropland can be fully compensated through extensification of 16.4 Mha of pure cropland when switching to the crops with the highest calorie supply (E-HC), or 30.9 Mha with the highest suitability crops (E-HS). Typically, high-suitability crops have lower calorie yields than the high-calorie crops, which explains the difference in the estimated areas transformed. This extensification mainly occurs in southern France, Hungary and Slovakia, primarily involving maize or wheat cultivation (Supplementary Figs. 7, 8). Such cropland transformations are constrained by the requirement to preserve current productions of vegetables, fruits and olives, which are relatively common cultivations in southern Europe (e.g., Spain and Italy). Consequently, these regions show minimal extensification, despite being affected by water scarcity and encompassing some biodiversity hotspots. Overall, both intensification and extensification scenarios rely on increased production of maize or wheat, and other crops generally show varying degrees of reduction.

Some of the proposed crop-switching strategies can increase the supply of certain macrominerals and micronutrients (mainly sodium, magnesium, zinc, and selenium), while causing declines in protein (up to −3.83% in E-HC) and other nutrients (e.g., around −10% for Selenium in E-HC and E-HS) (Supplementary Table 2). However, cereals, root and oil crops—the main crops involved in switching—contribute generally less than 30% to daily intakes of protein, macrominerals, and micronutrients in Europe (Supplementary Table 3). These nutrients are primarily sourced from animal-based foods and nutrient-dense crops (pulses, vegetables, fruits and nuts). In our analysis, these nutrient-dense crops are excluded from crop switching to preserve their supply of key nutrients. The identified remaining shortages can be addressed through relatively small adjustments in crop supply and dietary patterns or supporting the deployment of nutrient-rich fruit trees in the extensification scenarios.

**Impact on the land-climate-biodiversity nexus**

A balance of 30-year annual average carbon sequestration and emission flows indicates that all investigated scenarios have a potential for net carbon sequestration of 62–223 Mt $CO_2$-eq. $yr^{-1}$ (Fig. 5),

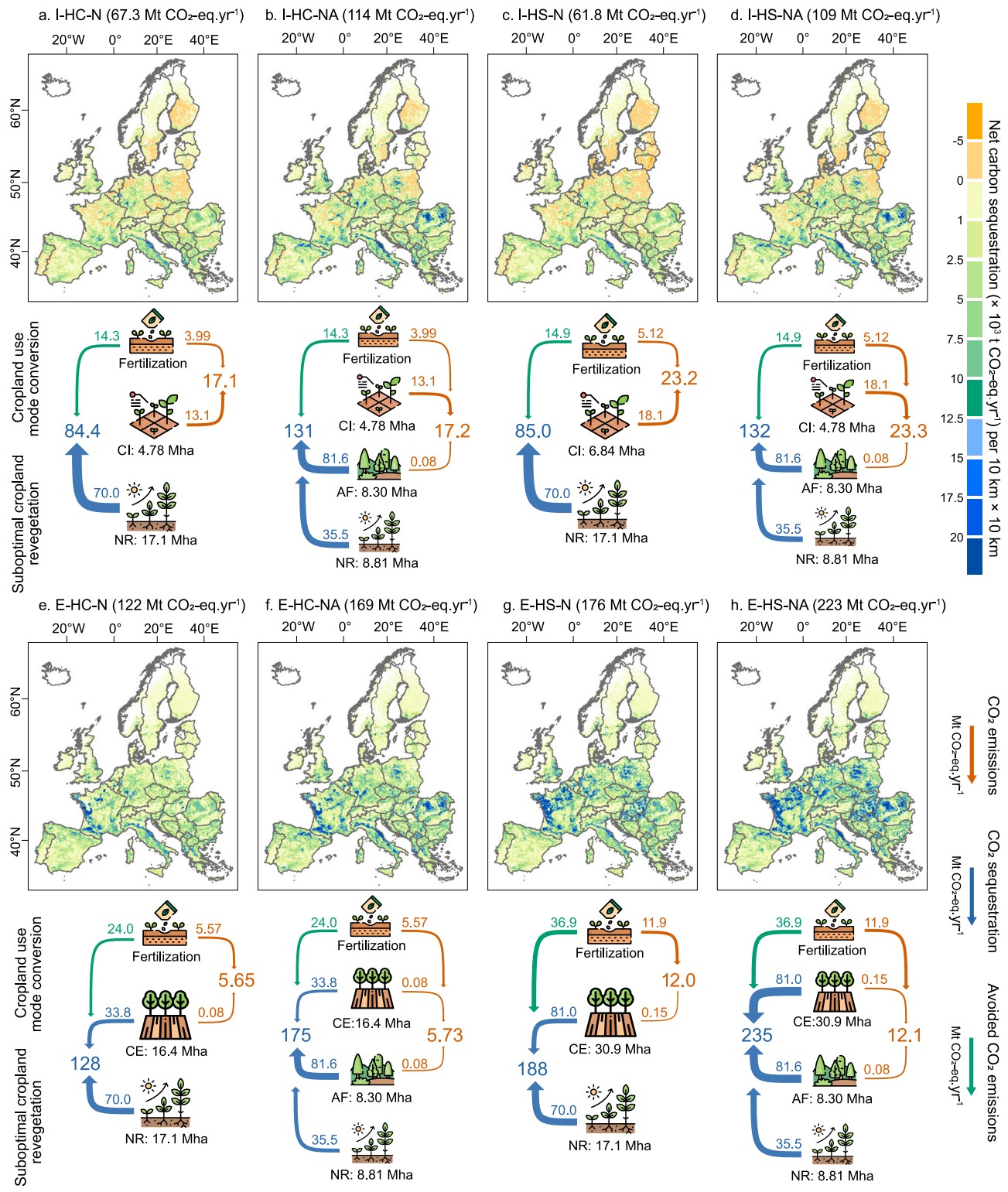

corresponding to 13–48% of the EU's agriculture sector's GHG emissions (467 Mt $CO_2$-eq.) in 2022[64]. This estimate accounts for revegetation of suboptimal cropland, vegetation changes following intensification or extensification, and shifts in soil $N_2O$ emissions associated with fertilizer use. The lower potential is achieved with the intensification scenarios when natural vegetation regrowth is the only option considered for suboptimal cropland (i.e., I-HC-N and I-HS-N). In this case, there is a relatively small difference induced by switching to the crop with the highest calorie supply or suitability (67 vs. 62 Mt $CO_2$-eq. yr$^{-1}$). An additional carbon sequestration benefit of 47.1 Mt $CO_2$-eq.

yr$^{-1}$ is achievable if afforestation is implemented on suboptimal cropland outside areas of biodiversity priority and water scarcity (i.e., I-HC-NA and I-HS-NA). The revegetation of suboptimal cropland can avoid fertilization-induced emissions of 12.7 Mt $CO_2$-eq. yr$^{-1}$ (Supplementary Fig. 11). In the intensification scenarios, the original crops in the crop-switching regions induced fertilization-related emissions of 1.59–2.19 Mt $CO_2$-eq. yr$^{-1}$, while after switching to the crop with the highest calorie supply or suitable soil, $N_2O$ emissions become 3.99–5.12 Mt $CO_2$-eq. yr$^{-1}$. The carbon emissions from clearing vegetation in mosaic cropland (13.1–18.1 Mt $CO_2$-eq. yr$^{-1}$) reduce the overall

**Fig. 5 | Carbon sequestration and emissions for the eight investigated scenarios combining cropland use mode with crop switching strategies and revegetation options.** The data correspond to carbon fluxes averaged over 30 years. Maps (**a**–**h**) show the net carbon sequestration, aggregated to 10 km resolution from 300 m for better visualization. The corresponding flow charts in (**a**–**h**) illustrate the carbon emissions (orange), sequestration (blue) and avoided emissions from suboptimal cropland and crop-switching (green). The values of the arrows represent the total outcome of the different processes, including cropland intensification or extensification, fertilization reduction (green) and increase (orange), afforestation (indirect emissions for tree planting) and natural regrowth. The net carbon sequestration of each scenario is labeled next to the subfigure's title. For example, in (**a**) 13.1 Mt (million tonnes) $CO_2$-eq. $yr^{-1}$ are emitted from clearing non-crops vegetation when converting 4.78 Mha (million hectares) of mosaic cropland to pure cropland, and 70.0 Mt $CO_2$-eq. $yr^{-1}$ are sequestered from revegetation of 17.1 Mha of suboptimal cropland via natural regrowth; at the same time, 14.3 Mt $CO_2$-eq. $yr^{-1}$ are avoided from reduction in fertilizer use, while 3.99 Mt $CO_2$-eq $yr^{-1}$ are emitted from increased soil $N_2O$ emissions following crop switching. The total emissions are thus 17.1 Mt $CO_2$-eq. $yr^{-1}$ and the total carbon savings are 84.4 Mt $CO_2$-eq $yr^{-1}$, resulting in a net sequestration of 67.3 Mt $CO_2$-eq. $yr^{-1}$. I: cropland intensification; E: cropland extensification; HC: crops with the highest calories supply; HS: crops with the highest suitability; N: natural vegetation regrowth; NA: combination of natural regrowth and afforestation, with afforestation only allowed in regions outside biodiversity priority and water scarcity areas. Locations of natural regrowth and afforestation of suboptimal cropland are shown in Supplementary Fig. 5, the location of cropland intensification and extensification in Fig. 4, and separate maps of avoided carbon emissions, carbon emissions, carbon sequestration, and uncertainty ranges are shown in Supplementary Figs. 11–14. Basemaps in (**a**–**h**) are sourced from GADM, with Austrian data licensed under Creative Commons Attribution-ShareAlike 2.0 (source: Government of Austria). Icons are obtained from Flaticon.com.

benefits of intensification (Supplementary Fig. 12). Local outcomes are determined by the spatial overlap of suboptimal cropland and mosaic cropland subject to intensification. In parts of northern Europe, net emissions occur because those from land clearing exceed sequestration from revegetating suboptimal cropland. This implies that intensification-based strategies, despite yielding overall climate benefits at a European level, may result in net emission increases in some countries, which is at odds with national climate change mitigation goals. Larger mitigation potentials (consistently achieved throughout the European countries) are observed in the extensification scenarios, where carbon sequestration from the new vegetation in mosaic cropland adds to that from revegetating suboptimal cropland, with relatively minor emissions from afforestation activities (Supplementary Fig. 13). In addition, abandoning cultivation of crops in the cropland areas subject to extensification avoids 11.3–24.1 Mt $CO_2$-eq. $yr^{-1}$ of soil $N_2O$ emissions, whereas the introduction of new crops following crop switching induces the emission of 5.57–11.9 Mt $CO_2$-eq. $yr^{-1}$. Extensification with the switch to the most suitable crop and a combination of natural regrowth and afforestation (E-HS-NA) is the scenario with the highest mitigation potential (223 Mt $CO_2$-eq. $yr^{-1}$).

A comparison of the scenario performances under multiple indicators connected to carbon sequestration potential, changes in biodiversity pressure, crop supply, and total extent of pure and mosaic cropland in Europe is presented in Fig. 6. Except for I-HS-N and I-HS-NA, the other six scenarios can fully compensate for the calorie losses from revegetating suboptimal cropland. Extensification scenarios achieve more effective cropland savings (25.4–32.6 Mha vs. 13.8–14.8 Mha under intensification) but necessitate sustaining extensive artificial planting within mosaic cropland to maintain productivity gains. Intensification scenarios shrink the total area of cropland in mosaics from today's 16.4 Mha to about 9 Mha, whereas the extensification scenarios could increase it up to 27.7 Mha. All scenarios can reduce from 5.22% to 17.6% of the overall biodiversity pressure from cropland, quantified using a simplified approach based on differences in potential species richness under different land use types relative to natural vegetation (Supplementary Fig. 15)[54,55]. Higher reductions are achieved with extensification combined with natural vegetation regrowth in suboptimal cropland. Revegetation measures of suboptimal cropland are the main contributors to the biodiversity benefits, followed by expansion of mosaic cropland in the extensification scenarios (which alone could reduce biodiversity pressure by 3.79% to 7.21%) (Supplementary Fig. 16). In contrast, clearing mosaic cropland in the intensification scenarios contributes to increased pressure on biodiversity. Overall, some carbon-biodiversity trade-offs can be observed in the scenarios investigated. They are mainly connected to the higher carbon sequestration rates, but lower biodiversity benefits of afforestation, relative to natural regrowth. For example, I-HC-N relative to I-HC-NA can provide about 60% higher biodiversity benefits but about 40% lower carbon sequestration. Similar findings are

observed in other scenarios differing in revegetation options. Italy and Romania are among the countries that are most affected by these trade-offs, as they have the largest available areas for either natural regrowth or afforestation (Supplementary Fig. 5).

## Discussion

This study quantifies the potential for European cropland to reconcile crop production with climate change mitigation and biodiversity conservation. Environmental benefits of revegetating suboptimal cropland are evident for climate regulation, biodiversity conservation and land degradation mitigation. Transforming agricultural production in the remaining cropland via intensification, extensification, and crop switching strategies has the potential to compensate for the crop production losses from abandoning suboptimal cropland. All the scenarios analyzed have their respective advantages and disadvantages. Intensification reduces the climate and biodiversity benefits of revegetating suboptimal cropland because of the carbon emissions from clearing mosaic cropland, but it is the strategy that requires the lowest extent of land-use transitions (4.78 Mha, when switching to the crop with the highest calorie supply). When intensification is combined with the highest-suitability crop, intensifying 6.84 Mha of cropland available outside the predefined constraints (i.e., biodiversity priority areas, water scarcity, and land degradation) falls short of fully compensating for the production losses. This means that relying solely on the cultivation of locally dominant crops in crop switching is insufficient to maintain crop production volumes constant. Changing to either a crop type with higher calorie yields or expanding cultivation into vulnerable areas is required. Intensification typically results in high-density cropping systems that are more vulnerable to biotic stressors and climatic conditions, and are associated with substantial environmental impacts[20–22,65]. On the other hand, extensification scenarios can compensate for crop losses from revegetation of suboptimal cropland, but they require extensive land-use changes (between 16 and 31 Mha) and crop switching, as well as careful management of artificial planting within mosaics. However, this cropland mode change ultimately leads to a 50% increase in mosaic cropland area in Europe, with potential environmental benefits. Structurally complex landscapes that are a mix of trees and crops help moderate microclimatic conditions by buffering weather extremes, and enhance the population of pollinators and natural pest predators, which in turn alleviate abiotic and biotic stress on crops' growth[23,24]. The integration of trees also improves soil moisture and nutrient retention and enriches soil fertility through increased organic matter accumulation, thereby sustaining crop yields, harvest rate, and habitat diversity[25–27]. The benefits for climate change mitigation and biodiversity conservation increase accordingly, despite trade-offs between revegetation options. Selective afforestation can achieve higher carbon sequestration rates, while natural regrowth is more cost-effective and beneficial for habitat restoration[66]. Relative to

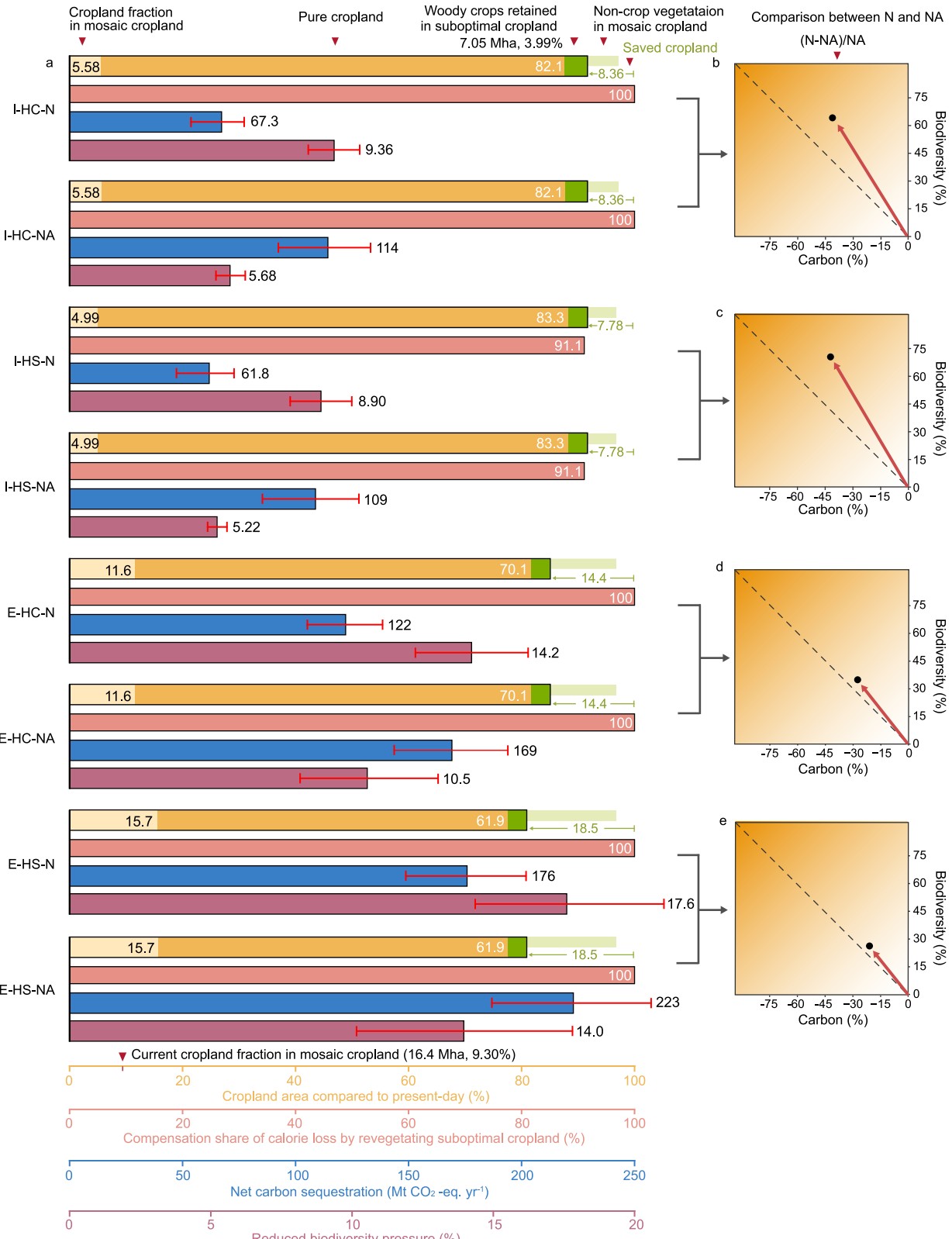

high-density cropland, artificial plantations also have positive effects for biodiversity, provided that a mix of native species is grown instead of monocultures[67,68].

Among the investigated scenarios, maize and wheat emerged as the most relevant crops, consistent with previous reports[69]. Maize typically delivers higher calories per unit area, while wheat is the preferred option in terms of local suitability. This approach represents a compromise between maximizing cropland-use efficiency and aligning with farmers' acceptance and local dietary preferences. Previous crop-switching studies often account for region-specific socioeconomic or environmental concerns, such as enhancing farm profitability in developing countries with high-return crops[28,70,71], reducing water demand in water-scarce regions through the introduction of drought-tolerant crop species[70,72], or improving climate resilience in

**Fig. 6 | Impact of the eight investigated scenarios on the land-climate-biodiversity nexus, combining cropland use mode with crop switching strategies and revegetation options.** Bar charts (**a**) show the impact of various solutions on the area of pure cropland and mosaic cropland (yellow, Mha: million hectares), compensation share of the calories lost by revegetating suboptimal cropland (pink), net carbon sequestration (blue, Mt: million tonnes), and reduced biodiversity pressure (purple). Each indicator refers to its corresponding colored horizontal axis at the bottom. The stacked yellow and green bar charts show the shares of mosaic cropland (light yellow), pure cropland (dark yellow), and woody crops retained in suboptimal cropland (dark green) of the total cropland in Europe after the implementation of cropland use mode conversion and crop switching strategies. The share of non-crop vegetation in mosaic cropland (light green bar) and saved cropland compared to present-day (green text) is also indicated next to the bars. The present-day mosaic cropland area (16.4 Mha, 9.3%) is indicated on the x-axis. The uncertainty ranges in the estimates of net carbon sequestration and biodiversity impact are included, and they represent the pairing of the lowest-bound with the highest-bound estimate from the use of the uncertainty factors in the original datasets. I: cropland intensification; E: cropland extensification; HC: crops with the highest calories supply; HS: crops with the highest suitability; N: natural regrowth; NA: combination of natural regrowth and afforestation. The panels on the right (**b**–**e**) show the potential trade-off between carbon and biodiversity of the natural regrowth option relative to the combination of natural regrowth and afforestation. For example, the top right figure indicates that intensification with crop switching to the one with the highest calorie supply provides 64.7% more biodiversity benefits but 41.1% less carbon sequestration benefits with natural regrowth than the combination of natural regrowth and afforestation (i.e., I-HC-N versus I-HC-NA).

agricultural intensive areas[73,74]. Some of these diverse aspects have been incorporated into the modeling framework of this study (e.g., water constraints), but the current high mechanization level of European agriculture, together with supportive policies (Supplementary Table 4), reduces the need to solely prioritize high-return crops and favor solutions that can also consider environmental aspects and integrated production systems.

In general, technical and socio-economic challenges increase with the spatial extent of the required changes. Abandonment of suboptimal cropland could inevitably affect local livelihoods and face issues of public acceptance, with possible consequences for local food supply. For example, all crops considered in the switching strategies are commonly grown in Europe, so the knowledge and technologies for their cultivation are largely available; however, changing crop types may be hindered by cultural and historical reasons. Adverse effects can be mitigated by financial compensation for cropland abandonment, educational campaigns and valuing natural revegetation measures. The implementation of mixed approaches of revegetation, intensification, and extensification at small scales can differ and be tailored to local priorities, while progressively guiding the collaboration among countries and a transition toward the intended direction. Acknowledging that spatial shifts in cropland areas are a dynamic process that is mainly guided by socio-economic drivers, rather than biophysical limitations of the land, offers an opportunity for a more sustainable governance of the patterns. In Europe, about 25 Mha of cropland has been abandoned since the early nineties[75], and another 20 Mha is at risk of abandonment by 2030[76], which is close to the estimate of suboptimal cropland achieved in this study. The EU Nature Restoration Regulation[77] mandates that at least 20% of the EU land is under restoration by 2030, and several countries are already implementing carbon- and biodiversity-focused reforestation programs of degraded or marginal cropland, indicating the practical feasibility of the revegetation potential investigated in this study (see Supplementary Table 4 for an overview of existing country-level policies that support revegetation of suboptimal cropland). Further, the EU farming policy is increasingly integrated with climate and environmental goals in recognition of the value of multifunctional agricultural landscapes. For example, the EGD[33] and the Biodiversity Strategy for 2030[34] promote the restoration of high-diversity landscape features, such as buffer strips and agroforestry plots, as integral components of climate adaptation and biodiversity conservation goals. Under the 2023–2027 national CAP strategic plans[35], there are financial mechanisms in support of agri-environment-climate measures. These plans, which must align with EU-wide sustainability objectives, generally have a joint focus on strengthening farmers' incomes, ensuring food security, and protecting the environment and climate. Results in this study are consistent with these perspectives, and they can be instrumental to further improve the effectiveness of the next round of national CAP strategic plans.

This study depends on a range of assumptions, uncertainties and limitations that are discussed in detail in the dedicated uncertainty section (Supplementary Text 3). A quantitative uncertainty analysis has been conducted on how variability in key parameters can affect estimates of suboptimal cropland area, how crop yield fluctuations can impact the area required for cropland intensification and extensification, and on the sensitivity of climate change mitigation benefits and biodiversity impacts to the datasets used to model carbon fluxes and species richness. Briefly, the identification of suboptimal cropland is mostly sensitive to the threshold used for the slope of the terrain. The total estimates can range from 20.6 to 29.5 Mha, against 24.2 Mha reported in the main results (Supplementary Table 5), but their spatial distribution patterns (Supplementary Figs. 17, 18) are broadly consistent with those presented in the main text (Fig. 2). Similarly, yield fluctuations can induce variability in the estimates of total crop production in suboptimal cropland that range from 55.8 Pcal to 69.2 Pcal, against 61.3 Pcal reported in the main results, with corresponding changes in the required areas used for cropland intensification or extensification (see Supplementary Fig. 19). Climate change mitigation potentials and biodiversity impacts have been computed considering the underlining uncertainty in the datasets used, and the associated uncertainty ranges are shown in Fig. 6. The results have been also compared with previous studies. The comparison has been conducted for the individual key components of our analysis, such as estimates of European suboptimal cropland area (Supplementary Table 6), carbon sequestration rates of natural regrowth (Supplementary Table 7) and afforestation (Supplementary Table 8), species richness difference between cropland and natural ecosystems (Supplementary Table 9), and impact on crop yields of agricultural extensification (Supplementary Table 1). Our overarching interpretation is that estimates of individual effects and drivers are in line with previous studies for similar study areas, and that our results are conservative, especially in terms of the climate change mitigation benefits that can be expected.

One main limitation of the analysis is that possible changes in crop production from high-yield farming due to climate change are not considered. Securing high yields over time requires the timely implementation of climate change adaptation strategies, especially in highly productive cropland, which is not explored in our analysis. Lack of adaptation measures can result in crop yield declines and affect the total area of cropland needed to sustain agricultural production or increase reliance on food imports. Future analysis can further embed the main conclusions of this study within a context of climate change adaptation designed around the implementation of alternative measures and different possible warming scenarios. Our analysis does not distinguish crop production for humans or livestock, and it does not include the influence of possible changes in dietary patterns towards less consumption of animal-based products (e.g., meat and dairy products). Changes in diets are an effective option to reduce agricultural land use, thereby enabling alternative uses of the land for climate change mitigation and biodiversity conservation[78,79]. In

Europe, about 60% of the gross production of cereals is used for animal feed[30], but only 15% of the total feedstuffs is actually in competition with food, since livestock primarily consumes roughages (e.g., grass and hay) and crop processing residues that are not typically consumed by humans[80]. Transitioning to more plant-based diets still faces a variety of social, technical, and educational barriers, but it has the potential to reduce the demand for agricultural land and facilitate the implementation of the investigated scenarios.

Overall, our study indicates that current farming practices and stakeholder engagement, instead of biophysical limitations of the land, hinder the design of more effective and sustainable agricultural systems in Europe. European policies have traditionally placed strong emphasis on agricultural production, but more holistic approaches and efforts towards diversified cropland landscapes that consider multiple environmental dimensions start to emerge. Proposals for implementing actions based on existing knowledge and for further improving policy frameworks already exist. The introduction of a GHG price can give resources to finance a sustainable transition in the agricultural production sector[81,82]. Adopting a cost-effective carbon price of 100 US$ per t $CO_2$[83], could yield an estimated annual profit between 6–22 billion US$ through revegetation of suboptimal cropland and extensification efforts. This fund can support a win-win and sustainable transformation of the European agri-food sector, provided that governments, consumers and farmers are engaged in a coordinated effort.

## Methods

### Mapping suboptimal cropland

Cropland located on steep slopes, fragmented or isolated, and with relatively low agricultural productivity, is identified as suboptimal. This land is typically characterized by low economic value, difficulties in agricultural mechanization, high fertilizer requirements, risks of unsustainable long-term yields and high environmental impacts[39,84]. Other degradation processes or unfavorable conditions (e.g., high soil erosion, depletion in soil organic matter or nutrients) are excluded from the criteria used to identify suboptimal cropland, because mitigation measures such as mulching, windbreaks, cover crops, biochar, etc., can be applied to restore the land. These land degradation threats are instead used to assess the extent to which they affect the identified suboptimal cropland.

We used the European Space Agency−Climate Change Initiative (ESA-CCI) land cover product[85] (resolution: 300 m at the equator) to map European cropland, defined as land primarily dedicated to the cultivation of herbaceous and woody crops under irrigation, rainfed, or post-flooding, excluding pasture and managed grassland[85]. This dataset was specifically developed to address the limitations of previous land cover datasets and can realistically represent land dynamics[86]. It provides annual timeseries of land cover from 1992 to present and include six cropland classes in classification system: 10 (Cropland, rainfed), 11 (Cropland, herbaceous cover), 12 (Cropland, tree or shrub cover), 20 (Cropland, irrigated or post-flooding), 30 (Mosaic cropland (> 50%)/natural vegetation (tree, shrub, herbaceous cover) (< 50%)), and 40 (Mosaic cropland (< 50%)/natural vegetation (> 50%)). In this analysis, classes 10, 11, 12, and 20 were reclassified as pure cropland (or high-density cropland), while classes 30 and 40 were reclassified as mosaic cropland (or low-density cropland, where cropland is mixed with non-crop vegetation cover). The robustness of the cropland classification has been validated in previous studies and shows a global user's and producer's accuracies of 81% and 92% (median), with an overall accuracy of 83%[87]. The particularly high classification accuracies make this product well-suited for cropland monitoring.

The cropland (pure and mosaic cropland pixels from the 2022 land cover data) located on high slope terrains (i.e., slope cropland) is identified using the Multi-Error-Removed Improved Terrain digital elevation model (MERIT DEM)[88], which was developed to correct Shuttle Radar Topography Mission (SRTM) for absolute elevation biases using high-precision Light Detection and Ranging elevation from Ice, Cloud and Land Elevation Satellite (ICESat) and ancillary datasets as references. Cropland is classified as suboptimal if the slope exceeds 8°, an average threshold for defining steep slope for cropland in a range of reviewed studies (4.6°–11.3°, Supplementary Table 10).

Small and highly fragmented crop parcels hinder agricultural mechanization and advanced practices, while also generally increasing transport distances in the food supply chain[84,89]. Cropland pixels, either pure or mosaic cropland, are classified as fragmented when they are without other nearby pure (or mosaic) cropland pixels (Supplementary Fig. 20). Because 94% of individual cropland plots in high-income countries exceed 10 ha[84], the use of a single ESA-CCI pixel is consistent with the typical field size, given that one pixel approximates 10 ha.

Low-productive cropland is identified with a method based on the growing season's cumulative normalized difference vegetation index (NDVI) as a proxy for agricultural productivity across 164 main sub-regions. The z-score method is used to determine the low-productivity threshold in each subregion, with z-scores ranging from −0.5 to −2 (normally distributed centered at −1.25) assigned based on the sub-region's ranking in cropland relative abundance (where lower abundance corresponds to a z-score closer to 0). This aims to identify a higher proportion of low-productive cropland in subregions with unfavorable farming conditions, and vice versa (see Supplementary Text 1 for methodology details, and Supplementary Fig. 21 for a schematic description of the approach).

In the uncertainty analysis, alternative thresholds of slope, fragmentation and productivity were used to estimate possible variations in the identified area of suboptimal cropland (as summarized in Supplementary Table 5). Specifically, 9° and 7° were tested for slope, a size of 10 ha or 20 ha was used for the identification of isolated patches of fragmented cropland, and a range of z-score from −0.75 to −2 (normal distribution centered at −1.375) and from −0.25 to −2 (normal distribution centered at −1.125) assigned across subregions was considered for low-productivity cropland, representing the lower- and upper-bound estimations, respectively. A minimum slope gradient of 7° is adopted as it corresponds to the minimum standard used in previous European research (as shown in Supplementary Table 10). The threshold of 10 ha approximately aligns with the minimum detectable patch size in the ESA-CCI land cover dataset. The results of the uncertainty analysis are discussed in detail in the dedicated uncertainty section (Supplementary Text 3).

### Scenario description

The scenario design integrates three main components: (1) revegetation options for suboptimal cropland (natural regrowth or afforestation); (2) adjustments in cropland use modes on remaining non-suboptimal cropland (cropland intensification or extensification); and (3) crop switching strategies (selecting either the crop with the highest calorie supply or the one with the highest suitability). These components are combined into eight scenarios (Fig. 1), which aim to compensate for crop production losses resulting from the revegetation of suboptimal cropland. In the intensification scenarios, crop production losses are compensated by expanding cropland in vegetation-cropland mosaic pixels and crop switching. In the extensification scenarios, parcels of trees are introduced into pure cropland pixels. The crop production losses from revegetating suboptimal cropland and from cropland area reduction due to extensification are compensated by crop switching and the relative yield benefits from landscape mosaicization (+9.5% on average, see Supplementary Text 2, and Supplementary Figs. 10, 22 for methodology details). Various thresholds for land-use constraints are considered, including: 1) exclusion of cropland intensification in areas of biodiversity priority[51], land degradation

risk[38] or water scarcity[52], wherein cropland extensification is prioritized; and 2) exclusion of afforestation in areas of biodiversity priority or water scarcity, wherein natural regrowth is prioritized. Further details on the individual components of these scenarios are provided in the subsequent sections.

## Revegetation options of suboptimal cropland

Of the total identified suboptimal cropland (24.2 Mha), 7.05 Mha of land for woody perennial crops (e.g., olive and fruit trees) is retained, and the rest is revegetated through either natural regrowth or active afforestation. Natural regrowth involves passive recovery towards forest biomes with over 25% forest cover[48], while afforestation entails artificial plantation practices that potentially enhance carbon sequestration rates relative to natural regrowth. Two scenarios of revegetation options are considered: 1) natural regrowth across all suboptimal cropland (17.1 Mha); and 2) natural regrowth on suboptimal cropland located within biodiversity priority and water-scarce areas (8.81 Mha), with afforestation applied to the remaining land (8.30 Mha, Supplementary Fig. 5). Biodiversity priority areas were defined as top 30% areas of global importance for conserving terrestrial biodiversity, considering terrestrial vertebrates, vascular plants, representative biomes, and existing protected areas, in line with the goals of the Global Biodiversity Framework[51]. Water scarcity refers to agricultural areas affected by blue water scarcity[52], where irrigation is unsustainable and the availability of renewable fresh water in surface and groundwater bodies available for human use is insufficient to sustainably meet crop water requirements. The distribution of areas dedicated to natural regrowth or afforestation greatly varies among countries (Supplementary Fig. 5). For instance, water scarcity is a main constraint for afforestation in arid regions like Spain, while biodiversity priority areas limit the extent of artificial planting in Greece and other Balkan countries.

## Crop production losses by revegetating suboptimal cropland

Revegetation of suboptimal cropland entails a loss in crop production. GAEZ + 2015 database with 5 arc-min resolution, including crop production, yield and harvested area, was used to estimate the crop production losses[50]. The GAEZ + 2015 database combines national and sub-national crop statistics with the support of detailed spatial multi-source data, including irrigation, soil properties, climate, human activity, and observed crop phenology and crop calendars. The GAEZ + 2015 database has been widely applied in agricultural studies[36,69,90], and is in agreement with other yield models and regional crop statistics[50]. Another commonly used crop yield database the Spatial Production Allocation Model (SPAM 2020)[58] with 5 arc-min resolution, is used as a sensitivity test for benchmarking crop production in suboptimal cropland. This database integrates crop statistics, expert-elicitation prior probabilities of crop distribution, crop-specific suitability and a cross-entropy approach to spatially estimate the crop distribution and yield. The validation based on independent datasets indicates a relatively high reliability with an $R^2$ from 0.71 to 0.91 for different crops[59,60]. Both GAEZ + 2015 and SPAM 2020 rely on FAOSTAT (sub)national data to provide control totals for crop area and yields, but differ in data sources, downscaling method, and irrigation mapping[59,60].

As GAEZ + 2015 database provides crop production data at 5 arc-min resolution, and the ESA-CCI land cover dataset used to identify suboptimal cropland is at ~300 m resolution, we used the ESA-CCI land cover map for the year 2022 as a land-cover base layer and reallocated the crop production from GAEZ + 2015 to a finer ESA-CCI pixel scale, following the approach from previous studies[36,60,69,90]. The same crop productivity was assumed for all types of croplands (whether it is suboptimal or not) within the same GAEZ + 2015 pixel, following refs. 69,90. In suboptimal cropland, all cereal, oil, root and sugar crops, as well as vegetables and pulses

were cleared, while woody crops were retained. GAEZ + 2015 provides production data measured in units of mass (tones) per pixel for cereal, oil, sugar and root crops. In contrast, a consistent set of international price weights compiled by FAO (in US dollars) was used instead of mass-based data to represent the production of vegetables and pulses[91], for which harvested area is reported. We assumed the calorie yield of vegetables and pulses per pixel to be 10% of the local cereal, oil, sugar and root crops, based on their relative calorie contribution in the reference diets and crop structure[92]. In the revegetation of suboptimal cropland, the calorie supply from vegetables and pulses was lost (only woody crops are retained, e.g., olives and fruit trees), while that from areas outside suboptimal cropland was retained and excluded from intensification and extensification scenarios. Harvest-weight production from GAEZ + 2015 was aggregated to consumer-level dry-weight production and converted into calorie productivity, using FAO food balance sheets[93], conversion factors between harvest-to-dry-weight[91], and food waste (17%)[69,94] and crop loss rate[95] during production, supply chain, and consumption stage. The total calorie supply of each 5 arc-min pixel (same as SPAM/GAEZ + 2015 database) was then calculated. This approach for calorie calculation is also applied to estimate calorie loss and supply in the crop switching strategies.

GAEZ + 2015 reports harvested area of crops, but it does not include areas where crops were planted but not harvested due to issues such as damage or crop failure[96]. It also omits fallow land within crop rotation schemes and space occupied by auxiliary cultivation infrastructure within the cropland pixel. Following the refs. 36,60,97, we used cultivation structure in cropping system (i.e., the relative share of crop area) per 5-min pixel reported by GAEZ + 2015 and multiplied it by the corresponding cropland extent from ESA-CCI data as a basemap to estimate the area of crop-specific cultivation process (e.g., vegetables and woody crops). According to the ESA-CCI dataset, mosaic cropland is assumed to consist of 50% cropland and 50% non-crop vegetation. This calculation is used to estimate the area to be retained from the revegetation of suboptimal cropland and crop switching, as well as the area available for crop switching in the intensification or extensification scenarios.

Compared to pure cropland, the presence of non-crop vegetation (i.e., multifunctional landscapes) affects the crop yields in the mosaic cropland because the presence of woody vegetation can regulate the microclimate, mitigate weather extremes, improve pest control and retain soil water and nutrient content, with growing evidence of cascading benefits for soil productivity and long-term yields[23,24] (see Supplementary Table 1). We estimated the relative impact of non-crop vegetation within mosaic cropland on yields in each subregion by comparing the yields between pure and mosaic cropland (Supplementary Text 2 for methodology details, and Supplementary Fig. 22 for a schematic description of the approach). It showed an average relative yield increase of 9.5% from the mosaicization of the landscape (Supplementary Fig. 10).

The calorie losses caused by revegetating suboptimal cropland are thus the following:

$$Cal\ loss_{5\,min} = \frac{50\% \times (1+\omega) \times N_{SM} + N_{SP}}{50\% \times (1+\omega) \times N_M + N_P} \times Total\ Cal_{5\,min} \quad (1)$$

*where Cal loss$_{5min}$* is the total calorie loss for each 5 arc-min pixel caused by revegetating suboptimal cropland. $N_M$ and $N_P$ are the pixel numbers of mosaic cropland and pure cropland based on ESA-CCI data at each 5 arc-min. $N_{SM}$ and $N_{SP}$ is the pixel numbers of mosaic suboptimal cropland and pure suboptimal cropland based on ESA-CCI data in the corresponding 5 arc-min pixel. $\omega$ is the relative impact of non-crop vegetation within mosaic cropland on yields. 50% indicates that the cropland area within a mosaic cropland pixel is half of the original pure cropland pixel. *Total Cal$_{5min}$* is the total calorie supply of

cereal crops, oil crops, sugar crops, root crops, vegetables and pulses for each 5 arc-min pixel.

## Cropland intensification

In this study, cropland intensification is implemented by converting mosaic cropland into pure cropland and by crop switching to either the crop type with the highest calorie supply or to the one with the highest local suitability (Supplementary Fig. 6). Another common approach for intensification is to increase local nitrogen-fertilizer input and nitrogen-use efficiency to close yield gaps. As yield gaps in Europe are relatively smaller than the global average, the achievement of higher yields is expected to require a consistent amount of additional nitrogen fertilizers (4.8 Mt), with adverse effects on soil GHG emissions and other environmental concerns[30]. This option is thus not considered in our analysis.

The required area for intensification is determined by the need to supply sufficient crops to offset the calories lost from revegetating suboptimal cropland. Conversion priorities are based on the neighboring pure cropland density (NPCD) of existing mosaic cropland, ranked from high to low. NPCD is defined as the number of pure cropland density adjacent to a central mosaic cropland pixel, identified using a moving window of $13 \times 13$ pixels (about 4 km $\times$ 4 km). A sensitivity test spanning window sizes from $5 \times 5$ to $21 \times 21$ pixels confirmed the robustness of this parameter, as all the resulting land areas show similar total values and distribution patterns (Supplementary Fig. 23). Within the same NPCD, conversion is prioritized in areas with higher net calorie supply potential after crop switching and applied only to pixels that yield net calorie gains. As high-density cultivation may exacerbate water consumption in areas affected by water scarcity, increase land degradation (e.g., soil erosion and SOC depletion) and negatively impact biodiversity, intensification is excluded from biodiversity priority areas[51], areas affected by water scarcity[52] and at land degradation risk. Additionally, an elevation threshold (<800 m) is applied to prevent agricultural intensification at high altitudes. Areas of biodiversity priority and water scarcity were described above. Areas under different forms of land degradation are sourced from Prăvălie et al.[38], which considered 12 types of degradation processes that challenge agricultural production in Europe, including water and wind erosion, SOC loss and soil pollution. At the pixel level, each degradation process is categorized as either "Non-critical" or "Critical", based on documented evidence that can trigger reductions/losses in agricultural land productivity or soil health. Areas at risk of land degradation are defined as places that are threatened by at least two co-occurring "Critical" land degradation processes following ref. 38.

Once the mosaic cropland pixels selected for intensification are identified, they are converted into new pure cropland pixels. The shares of vegetables, fruits, nuts, olives and pulses within the cropland fraction of the original mosaic cropland were retained to maintain the supply of key nutrients and preserve existing vegetation carbon stocks. The non-cropland fraction and other crops (e.g., cereal, sugar, root and oil crops) were replaced by the crops with the highest calorie supply or the highest suitability, as described below. The cropland not selected for intensification retains its current crop production.

## Cropland extensification

Cropland extensification refers to converting pure cropland to mosaic cropland, where half of the pixel is allocated to crops and the other half to non-crop vegetation. It aims to create multifunctional agriculture landscapes and mitigate the environmental impacts of intensive cultivation. The balance of crop losses from revegetating suboptimal cropland is expected to be offset by yield effects driven by multifunctional agriculture landscapes (while accounting for the reduction in available cultivation areas per pixel), along with crop switching to varieties with the highest calorie supply or highest suitability. The pure

cropland pixels that are prioritized for the conversion are those with high biodiversity value and at risk of land degradation (i.e., affected by 2 or more co-occurring degradation threats) and water scarcity, so as to reduce the overall environmental impact of agriculture. Prioritization criteria for cropland extensification are implemented according to the following orders: 1) pure cropland located in areas simultaneously identified as biodiversity priority, land degradation risk, and water scarcity; 2) within biodiversity priority areas and at risk of water scarcity; 3) within biodiversity priority areas and affected by land degradation; 4) within areas at risk of water scarcity and land degradation; 5) within biodiversity priority areas; 6) within water scarcity areas; 7) within areas affected by land degradation; and 8) the remaining pure cropland. Within each of these criteria, the pixels prioritized for the conversion are those with higher NPCD, and, where NPCD is equal, higher net calorie supply potential. Such conversion is restricted to pixels yielding net calorie gains after crop switching. In these pixels, the original cultivation shares of vegetables, fruits, nuts, olives, and pulses are retained, while 50% of the pixel area is revegetated with non-crop vegetation (via active afforestation). The remaining crop fraction of the new mosaic pixel is planted with crops with the highest calorie supply or the highest suitability. The cropland not selected for extensification maintains its existing crop production.

## Crop switching strategies

Two crop switching strategies are considered: 1) crop with the highest calorie supply and 2) crop with the highest local suitability. The highest calorie crop is determined by selecting the crop with the highest yield and calorie content per unit mass (i.e., calorie yield) within each pixel. The crop with the highest suitability is the crop with the largest present-day harvested area within a pixel. This option accounts for current farming practices and local preferences for crop cultivation. In the intensification scenarios, the crop switching is applied to the new pure cropland pixels (i.e., both the cropland fraction and the former non-cropland fraction within the converted mosaic cropland). In the extensification scenarios, the crop switching is applied to the cropland fraction within the new mosaic cropland. In the pixels where the crop switching strategy is implemented, the shares of vegetables, pulses, fruits, nuts, and olives were retained. A method similar to that of Eq. (1) was used to calculate the calorie loss from clearing original crops (i.e., cereal, sugar, oil, and root crops) in crop switching. Calorie gains in both strategies are derived from newly cultivated crops selected from 11 major crops commonly grown in Europe: barley, groundnut, maize, millet, potato, rapeseed, rice, sorghum, soybean, sunflower, and wheat[11,69]. Yield data are sourced from the GAEZ + 2015 database, aligning with present-day rainfed and irrigation conditions, and the effect of non-crop vegetation on yield for extensification scenarios was adopted. It is assumed that the newly cultivated crops are fully harvested following the refs. 11,69,90. A sensitivity test assuming an 80% harvest rate to represent suboptimal growth or crop failure is performed in the uncertainty section (Supplementary Fig. 24). In addition, an uncertainty analysis was conducted to assess the impact of crop-specific yield fluctuation on the required areas for cropland intensification and extensification, to compensate for crop production losses from revegetating suboptimal cropland. We calculated crop-specific yield fluctuations (expressed as absolute percentage changes) by considering the differences in yields for each crop across each of the 164 subregions between 2010 and the present. These fluctuations at a subregional level are then used to infer the upper- and lower-bound yield variations for each pixel, within the corresponding subregion for the investigated crops. The corresponding areas required for cropland intensification or extensification are used as uncertainty ranges and discussed in the uncertainty section (Supplementary Text 3).

## Carbon sequestration from natural regrowth and afforestation

Effects on climate change mitigation are considered by calculating the amount of carbon sequestration in aboveground biomass for the revegetation of suboptimal cropland and the expansion of non-crop vegetation cover in the extensification scenarios. The accounting also includes carbon emissions from clearing vegetation cover in the intensification scenarios, life-cycle emissions from afforestation activities, as well as soil $N_2O$ emission changes induced by variation in fertilizer use in suboptimal cropland and crop-switching regions.

The carbon sequestration rate for natural vegetation regrowth is sourced from a spatially explicit dataset of 30 arc-second resolution reconstructed from observations[48]. It indicates the potential aboveground carbon accumulation rates for the first 30 years of natural forest regrowth. The mapping of this dataset is based on the ensemble model of 100 random-forest models and combined investigated data from 257 studies and 13112 georeferenced measurements of aboveground carbon accumulation with 66 environmental covariate layers. Aboveground biomass includes stem and branch biomass. Natural vegetation regrowth is only considered for suboptimal cropland, and not for the revegetation of the pure cropland pixels converted to mosaic cropland in extensification scenarios. The standard deviation across 100 random-forest model predictions per pixel is used in the uncertainty analysis[48].

The carbon sequestration potential from afforestation is estimated using the Global Forest Model (G4M), driven by multi-ensemble mean climate data under SSP2-RCP4.5 from the EC-Earth3 model[49]. G4M incorporates biomass data from FAO yield statistics and MODIS net primary productivity to parameterize increment functions, with the support of various local factors (e.g., soil and climate characteristics) and forest management practices (e.g., locally optimal tree species, rotation period, and thinning intensity). Mean annual increment was used to represent tree growth and carbon sequestration rate, averaged over 30 years to ensure consistency with the natural vegetation regrowth dataset. Natural regrowth data consider the aboveground stem and branch biomass, while G4M simulates only the aboveground stem component; thus, a default standard factor of 20% is used to estimate the total carbon sequestration for afforestation[66]. G4M simulates growth rates for both coniferous and non-coniferous trees using local factors of soil fertility and climatic conditions. The tree species that can deliver the highest carbon sequestration rate at each pixel was considered for artificial plantations on suboptimal cropland. The resulting distribution of tree species on suboptimal cropland is 45.2% coniferous (3.75 Mha) and 54.8% non-coniferous (4.55 Mha) (Supplementary Fig. 25). At a European level, non-coniferous species are mostly located in the southern part of the domain, while the other species are more common in colder climates. Owing to a lack of uncertainty ranges from the original data, a default range of 15% of the pixel-scale estimated value is considered on the basis of the uncertainty level of natural regrowth data ($\pm 13\%$ of the predicted value on average).

The carbon sequestration of non-crop vegetation cover in mosaic cropland pixels was calculated as follows. In line with the ESA-CCI data classification, the mosaic cropland is considered to be 50% cropland and 50% non-crop vegetation (grassland or trees). We used the Global Forest Watch data[98] at a 30-meter resolution to calculate the average tree cover of the non-cropland share within mosaic cropland for each subregion. We assumed that the new mosaic cropland, resulting from the conversion of pure cropland pixels, has the same tree cover as the existing mosaic cropland pixels in each subregion and that non-coniferous species are planted. That is because non-coniferous species are typically better adaptive to local conditions than coniferous species, and have higher potential to mitigate the adverse effects of intensive agriculture activities[99,100]. Carbon sequestration rates from afforestation of mosaic cropland are estimated as the product of the G4M-based carbon sequestration value for non-coniferous species and the corresponding tree cover of the non-cropland share of the mosaic cropland.

Under the intensification scenarios, vegetation cover in the non-cropland fraction of mosaic pixels is cleared to expand cropland areas. The associated emissions were considered to originate from immediate oxidation. Such emissions are estimated by combining annual ESA-CCI land cover data with aboveground biomass carbon (AGBC) density map[101], using the widely adopted "space-for-time" method[102]. The mapping of the AGBC density data used a rule-based decision-tree approach to integrate various published biomass maps. This dataset has the same resolution as ESA-CCI data (300 m at the equator) and refers to the year 2010. For mosaic cropland classified as such in both 2010 and 2022, we used the corresponding AGBC value, while those newly identified in 2022 were given the mean AGBC of adjacent $13 \times 13$ (~ $4\,km \times 4\,km$) mosaic cropland pixels. In the case of afforestation, emissions from forestry operations (e.g., silviculture, seedling production, and tree planting) in suboptimal cropland and in mosaic cropland are accounted for, and they correspond to 0.30 t $CO_2$-eq. ha$^{-1}$, and 0.15 t $CO_2$-eq. ha$^{-1}$, which are both one-time emissions occurring when forest plantations are established[66]. The AGBC uncertainty layer used is based on the cumulative standard error propagated from multiple data sources used in the modeling process[101]. When results are shown in terms of annual $CO_2$ fluxes, these emissions are linearly distributed over a 30-year period to make them comparable with sequestration rates.

## $N_2O$ emissions from nitrogen fertilizer

Direct and indirect soil $N_2O$ emissions[53] associated with the application of synthetic nitrogen fertilizer and animal manure were quantified following the approach described in ref. 103. The calculation of $N_2O$ emissions considers the avoided emissions from abandoning agriculture in suboptimal cropland and the changes in $N_2O$ emissions induced by crop switching in the intensification/extensification scenarios. Crop-specific synthetic fertilizer application rates were obtained from the geospatial dataset NPKGRIDS[104], with 0.05° resolution. NPKGRIDS was developed using a data fusion approach to integrate the existing eight published datasets of fertilization, compiled from georeferenced survey data or national/subnational agricultural statistics. Validation against national/subnational levels demonstrated strong consistency with $R^2 = 0.80$. The manure application to croplands is sourced from the ref. 105, and was based on spatial-explicit estimates derived from livestock distribution density and corresponding manure-production rates.

The updated version of the NL-N-RR model[106,107] was used to estimate direct $N_2O$ emissions. This model (as shown in Eqs (2, 3)) is based on an exponential model trained by 985 measurements of $N_2O$ emission in agricultural fields extracted from 203 publications, outperforming traditional approaches, such as linear models and the IPCC method.

$$Y(x) = I(x) - I(0) \tag{2}$$

$$I(x) = \exp\left[\frac{\left((\mu_0 + \sigma_o^2)^2 - \mu_0^2\right)}{(2\sigma_o^2)}\right] \exp\left[\frac{\left((\mu_1 + \sigma_1^2 x)^2 - \mu_1^2\right)}{(2\sigma_1^2)}\right] \tag{3}$$

Where $Y$ is the crop-specific direct $N_2O$ emission rate (kg N ha$^{-1}$ yr$^{-1}$), $x$ is the crop-specific N-fertilizer rate (kg ha$^{-1}$), and the parameters of $\mu_O$, $\mu_1$, $\sigma_O$ and $\sigma_1$ are adopted from the ref. 106.

The estimates of indirect emissions were based on the 2006 IPCC Guidelines for National Greenhouse Gas Inventories[108] (Table 11.3 of Guidelines). These include N-application rate, the fraction of inputs volatized or leached, and IPCC default emission factors. We assume that leaching occurs when irrigation or when the difference between rainy season precipitation and rainy season potential

evapotranspiration is greater than the soil-water holding capacity[103]. For each pixel, the rainy season was defined as the months where precipitation exceeded one-third of the precipitation in the wettest month. Gridded monthly precipitation and potential evapotranspiration is obtained from the ERA5-land dataset[109], and soil-water holding capacity is sourced from the International Soil Reference and Information Center (ISRIC) soil data, named WISE30sec[110]. Impacts of $N_2O$ emissions are converted to $CO_2$-eq. using the Global Warming Potential for a time horizon of 100 years (GWP100)[111]. In the uncertainty analysis, a 15% range of the estimated value of $N_2O$ emissions at a pixel scale is considered.

### Biodiversity impacts and trade-off analysis

The biodiversity impacts of the different scenarios are estimated as the difference in local species richness between natural vegetation (used as a benchmark) and other land cover types. We used a 5-arc-min map of potential species richness[54,55] that describes the number of species within a given pixel of natural vegetation or pure cropland. This dataset incorporates species-specific extents of occurrence, habitat preferences, and tolerance to conversion from natural habitats to cropland, considering potential habitat range sizes of 16,919 mammal, bird and amphibian species. We integrated this species richness data with a meta-analysis comparing species richness differences across land-use types to infer the potential species richness supported by artificial plantations and mosaic cropland. Species richness in artificial plantations was estimated to be 71% (95% confidence interval: 68%–74%) of that in natural vegetation, as derived from a global meta-analysis based on 287 studies and 1008 matched data pairs[68]. Species richness in mosaic cropland was estimated to be 130% of the species richness in pure cropland. This coefficient is derived from a European biodiversity meta-analysis[26], based on 53 publications and 365 comparisons covering five principal European biogeographical regions. It showed that mixed landscapes of trees and cropland support 29.7% (95% confidence interval: 18.7%–40.7%) higher species richness compared to conventional cropland.

The biodiversity impact of current cropland ($BI_{CC}$) is quantified as the loss in species richness relative to a hypothetical conversion of all croplands to natural vegetation. It indicates the number of species estimated to be locally lost due to the conversion of natural habitat to pure or mosaic cropland. Then, we evaluated the biodiversity impact ($BI_{AC}$) after cropland use mode change (i.e., intensification or extensification) and revegetating suboptimal cropland (i.e., natural regrowth and afforestation). We assumed that suboptimal cropland after natural regrowth could attain the species richness supported by natural vegetation, while the species richness in the suboptimal cropland after afforestation corresponds to that of artificial plantations. The species richness of cropland after intensification and extensification corresponds to that of pure cropland and mosaic cropland, respectively. The cropland areas that are not involved in intensification, extensification, or revegetation retain their original species richness. Revegetation of suboptimal cropland by afforestation and natural regrowth, and increase of non-crop vegetation in high-density cropland by extensification, is predicted to improve local species richness, while the non-crop vegetation loss caused by intensification could decrease species richness. The difference between $BI_{CC}$ and $BI_{AC}$ represents the extent to which these investigated scenarios can alleviate biodiversity pressure from cropland. The lower and upper bounds of the 95% confidence intervals reported in the meta-analysis are used to apply the same procedure to the biodiversity-impact estimates, as uncertainty ranges. The resulting uncertainty bands are presented in Fig. 6.

Carbon–biodiversity trade-offs are assessed across scenarios with consistent cropland-use mode conversions and crop-switching strategies, differing only in the revegetation options for suboptimal cropland: full natural regrowth vs. natural regrowth combined with afforestation. The analysis compares the total carbon (via climate change mitigation) or biodiversity (through reducing biodiversity pressure from cropland) benefits of natural regrowth and intensification (or extensification) against a combination of natural regrowth, afforestation and intensification (or extensification), e.g., E-HC-N vs. E-HC-NA (see Supplementary Fig. 26 for a schematic of the calculation).

$$\Delta B_{c/b} = \frac{N_{c/b} - NA_{c/b}}{NA_{c/b}} \tag{4}$$

$\Delta B_c$ and $\Delta B_b$ are the difference in carbon and biodiversity benefits between two scenarios where cropland use mode and crop switching strategies are consistent while revegetation options differ. $N_c$ and $N_b$ represent the total carbon and biodiversity benefits, respectively, of scenarios relying solely on natural regrowth in suboptimal cropland, while $NA_c$ and $NA_b$ represent the corresponding benefits of scenarios combining natural regrowth and afforestation. Both $N_c$ and $NA_c$ account for carbon sequestration from revegetated suboptimal cropland (natural regrowth only for $N_c$; both natural regrowth and afforestation for $NA_c$), and trees established under extensification scenarios, emissions from land clearing and afforestation activities, as well as fertilization-induced $N_2O$ emissions change. Similarly, $N_b$ and $NA_b$ account for biodiversity pressure reduction from revegetated suboptimal cropland (natural regrowth only for $N_b$; both natural regrowth and afforestation for $NA_b$), trees established under extensification scenarios, and increased biodiversity pressure from cropland intensification.

### Reporting summary

Further information on research design is available in the Nature Portfolio Reporting Summary linked to this article.

## Data availability

All input data used in this study are publicly available from the corresponding references. Other data necessary to replicate the results can be collected from the following repositories: crop yield, production and harvested area from https://dataverse.harvard.edu/dataset.xhtml?persistentId=doi:10.7910/DVN/XGGJAV, and https://mapspam.info/, priority areas for biodiversity conservation from https://zenodo.org/record/5006332#.Y8RrOJjMI2w, and land degradation from https://esdac.jrc.ec.europa.eu/. The basemaps showing national boundaries are derived from the GADM database (https://gadm.org), with Austrian data licensed under Creative Commons Attribution-ShareAlike 2.0 (source: Government of Austria). Our modeling outputs are available at https://doi.org/10.5281/zenodo.16881974. Source data are provided with this paper.

## Code availability

Post-processing of model outputs and all subsequent calculations were performed in MATLAB and ArcGIS (https://www.arcgis.com/index.html). Visualizations were generated using the ggplot2 package in R and Adobe Illustrator (https://www.adobe.com/products/illustrator.html). The custom MATLAB scripts developed for this study are available at https://doi.org/10.5281/zenodo.16881974.

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

## Acknowledgments

We acknowledge the support of the Norwegian Research Council through the projects VOM (no. 319892) and MITISTRESS (no. 286773).

## Author contributions

T.H., X.H., and F.C. conceived and designed the study. T.H. performed and postprocessed the spatial analysis and drafted the manuscript. X.H. contributed with statistical analysis. G.A. and J.S. contributed with ecological interpretations. B.O. contributed to policy implications. F.C. provided detailed guidance throughout the process and wrote the manuscript. All authors contributed to interpreting the results and revising the manuscript.

## Funding

 Olavs Hospital - Trondheim University Hospital).

## Competing interests

The authors declare no competing interests.
