## [Transparent Peer Review file · Nature Communications]

Reconciling crop production, climate action and nature conservation in Europe by agricultural intensification and extensification

Corresponding Author: Dr Ting Hua

Version 0:

Reviewer comments:

Reviewer #1

(Remarks to the Author)

The manuscript contains numerous unclear definitions, including suboptimal cropland, revegetating cropland, and degradation risk, among others. These key terms need to be precisely defined to ensure clarity and consistency throughout the paper.

Additionally, to align with the paper's title, I would expect a quantified assessment of the benefits of agricultural intensification and extensification in terms of crop production, climate impact, and nature conservation. However, several logical inconsistencies appear in the abstract:

The authors discuss the amount of suboptimal cropland and its degradation risk, but it is unclear why this information is relevant in this context.

The claim of reducing intensity in suboptimal croplands is confusing—how is suboptimal cropland defined, and how does its classification relate to intensity reduction?

The abstract also contains excessive jargon, making it difficult to follow. Moreover, the relationships between crop optimization, revegetating cropland, and intensification are unclear—do these concepts all fall under intensification?

For the body of paper, the scenario settings are not well explained, making the methodology difficult to follow. Due to these significant issues in clarity, terminology, and logical coherence, I do not recommend this paper for publication in its current form.

Minor points:

1. In Figure 5, could you provide uncertainty estimates?
2. For all the simulated scenarios, how do you quantify the uncertainty range?
3. Regarding equation 5, how do you calculate $h_{Nc/b}$ and NAc/b ? Could you also provide a schematic figure for this part?

Reviewer #2

(Remarks to the Author)

The paper is highly relevant and presents novel and significant findings at a continental (European) scale. It focuses on the identification and mapping of suboptimal cropland, with the goal of revegetating these areas to improve environmental sustainability without compromising food security. This conversion would contribute to improving biodiversity, carbon sequestration, and climate change, which are spatially estimated, through scenarios addressing both intensification and extensification or crop switching strategies to compensate for the loss of production resulting from the transformation of these suboptimal agricultural areas.

The work makes a valuable contribution to the field by taking a significant step toward identifying solutions and scenarios that can achieve food security without major environmental compromise. This approach would contribute to aligning with the objectives set out in the European Green Deal for 2030, without compromising food security or increasing land demand

outside the EU. This vision is not only relevant but also urgent, given the current challenges of climate change and agricultural sustainability.

The central idea of the research is innovative and well-designed and executed. From a methodological standpoint, the approaches used, though I am not an expert in remote sensing, appear to be solid and well-founded. The methodology provides a robust basis for the replicability of the study. The results obtained clearly support the conclusions presented, suggesting that the study is consistent with its initial assumptions.

Although there is an important section on the uncertainties and limitations of the study, it would still be valuable to consider a deeper discussion of the potential challenges and limitations of the models used. This would offer a more comprehensive and nuanced view of the practical and policy implications of the conclusions. For example, the authors have not considered the potential impacts of intensification on biodiversity in situ on cropland. However, the differences between wildlife-friendly farming and farming intensification are important when considering biodiversity, and this distinction should be more clearly addressed in the study.

In summary, the findings of this work are deserving of publication in a prestigious journal such as Nature Communications (NC), given their value and relevance both to the scientific community and to policy and environmental decision-making. The quality and potential impact of this study could make it a notable contribution to the field of agricultural sustainability and climate change.

Reviewer #3

(Remarks to the Author)

This paper investigates how to reconcile agricultural production with climate action and biodiversity conservation in Europe by analyzing the potential for transforming suboptimal cropland. The authors identify 24.2 million hectares (Mha) of suboptimal cropland, which is underutilized and at risk of degradation, and evaluate different agricultural strategies (intensification and extensification) to preserve crop production while contributing to climate change mitigation and biodiversity conservation. The paper addresses a timely and critical issue regarding the balance between agricultural production, climate action, and biodiversity conservation through sustainable transformation.

Overall, the science presented is sound, and the paper offers original insights, particularly in the context of the ongoing push to achieve the Sustainable Development Goals (SDGs) by 2030. I read the manuscript with great interest and appreciate the depth of analysis. However, some explanations, particularly in the discussion, seem oversimplified. I recommend accepting the manuscript after major revisions.

General comments:

(1) The paper addresses multiple issues with highly complex influencing factors. Agricultural profitability involves different crop varieties and management practices. This paper incorporates environmental impacts, including biodiversity conservation, greenhouse gas emissions, and land protection. The interconnections, quantification, and trade-offs among these factors contribute to a high degree of uncertainty in the results.

(2) The paper comprehensively considers the interaction, or rather the compensatory relationship, between low-yield (suboptimal) farmland and high-yield farmland. For high-yield farmland, adaptive management for climate change needs to be implemented even in the absence of measures for low-yield farmland. Another possible scenario is that climate change mitigation subsidies could be provided directly to high-yield farmland managers, meaning there would be no interconnection between low-yield and high-yield farmland in this context.

(3) The paper primarily focuses on the scientific value of farmland management adjustments. It is highly relevant to understanding agricultural activities and their associated environmental protection measures. However, is it feasible from a policy perspective? The paper examines Europe as a whole, but for policy implementation, feasibility would depend on financial subsidies. Are agricultural and environmental policies independent in each European country? Does the EU framework provide a unified approach to environmental management and agricultural subsidies?

Specific comments:

It would be beneficial to compare the findings with previous studies (e.g., Rising and Devineni, 2020; Xie et al., 2023), particularly in terms of how they model crop-switching strategies for sustainability. This comparison would help position the current work within the context of existing research and highlight its novel contributions.

While the manuscript focuses primarily on calorie production, it is crucial to also consider the nutritional aspects of crop transformation (Denning, 2025). Achieving universal food security requires not only sufficient calories but also access to nutritious food to address malnutrition and micronutrient deficiencies. The paper mainly discusses caloric outputs, but it would be valuable to explore how nutrition is integrated into the crop-switching strategies. For example, Chakraborti et al. (2023) discuss how crop-switching can enhance environmental sustainability while improving nutrition and farmer livelihoods. I suggest adding a section that discusses how the proposed strategies can balance calorie production with the need for more nutritious crops.

The manuscript shows the potential for CO₂ emissions reduction, but I recommend that the authors also consider other significant greenhouse gases, such as nitrous oxide (N₂O). N₂O, especially from fertilizer application, is a potent greenhouse gas with much higher global warming potential than CO₂ (Carlson et al., 2016; Tian et al., 2020). Not incorporating N₂O emissions could lead to an underestimation of the true climate impact of the proposed strategies, and it could impact the assessment of their overall effectiveness in climate mitigation.

More discussion is needed on the mechanisms behind the results and how they align with past literature. For instance, how do agricultural intensification and extensification balance the trade-off between food production and climate change mitigation potential? This would help clarify how the proposed strategies work in practice and their potential implications for broader climate and food security goals.

I hope these comments contribute to improving the manuscript.

References:

- Carlson, K.M. et al., 2016. Greenhouse gas emissions intensity of global croplands. *Nature Climate Change*, 7(1): 63-68.
- Chakraborti, R. et al., 2023. Crop switching for water sustainability in India's food bowl yields co-benefits for food security and farmers' profits. *Nature Water*, 1(10): 864-878.
- Denning, G., 2025. Sustainable intensification of agriculture: the foundation for universal food security. *npj Sustainable Agriculture*, 3(1).
- Rising, J. and Devineni, N., 2020. Crop switching reduces agricultural losses from climate change in the United States by half under RCP 8.5. *Nat Commun*, 11(1): 4991.
- Tian, H. et al., 2020. A comprehensive quantification of global nitrous oxide sources and sinks. *Nature*, 586(7828): 248-256.
- Xie, W. et al., 2023. Crop switching can enhance environmental sustainability and farmer incomes in China. *Nature*.

Reviewer #4

(Remarks to the Author)

Reviewer #5

(Remarks to the Author)

Version 1:

Reviewer comments:

Reviewer #1

(Remarks to the Author)

The authors have made a clear effort to address my concerns. The revised text is clear, and I am satisfied with the revisions.

Reviewer #2

(Remarks to the Author)

The authors have adequately responded to all the reviewers' concerns and revised the manuscript accordingly. Now the manuscript is much stronger. I find the current version satisfactory and have no additional comments.

Reviewer #3

(Remarks to the Author)

The authors have carefully revised the manuscript based on the reviewers' comments and suggestions. The current version is satisfactory and publishable. I would recommend acceptance for publication.

Reviewer #4

(Remarks to the Author)

Reviewer #5

(Remarks to the Author)

REVIEWER COMMENTS

Reviewer #1 (Remarks to the Author):

1. The manuscript contains numerous unclear definitions, including suboptimal cropland, revegetating cropland, and degradation risk, among others. These key terms need to be precisely defined to ensure clarity and consistency throughout the paper.

>> We now try to better define these terms the first time they appear in the main text. Some examples:

- Suboptimal cropland is characterized by steep slopes, fragmented and isolated farms, and low agricultural productivity (lines 4-5, and 59-60, with more details provided in the methods section “Mapping suboptimal cropland”).
- Revegetating cropland refers to the process by which suboptimal cropland is converted to forested land or natural land cover by passive natural regrowth or active afforestation (lines 73-74).
- Degradation risk refers to a series of land degradation processes that negatively affecting land productivity and quality, as identified by Práválie et al. (2024) (lines 87-88). It includes water erosion, wind erosion, and soil pollution, among others. The detailed list of the land degradation processes is given in Fig. 3.

Reference:

Práválie, R., Borrelli, P., Panagos, P. *et al.* A unifying modelling of multiple land degradation pathways in Europe. *Nat Commun* **15**, 3862 (2024). <https://doi.org/10.1038/s41467-024-48252-x>

2. Additionally, to align with the paper’s title, I would expect a quantified assessment of the benefits of agricultural intensification and extensification in terms of crop production, climate impact, and nature conservation. However, several logical inconsistencies appear in the abstract: The authors discuss the amount of suboptimal cropland and its degradation risk, but it is unclear why this information is relevant in this context. The claim of reducing intensity in suboptimal croplands is confusing—how is suboptimal cropland defined, and how does its classification relate to intensity reduction? The abstract also contains excessive jargon, making it difficult to follow. Moreover, the relationships between crop optimization, revegetating cropland, and intensification are unclear—do these concepts all fall under intensification?

>> We thank the reviewer for these thoughtful comments, which have helped us improve the clarity of the abstract and manuscript. First, the benefits of agricultural intensification and extensification in terms of crop production, climate impact, and nature conservation are indeed the main goal of the paper, and they are

shown in Fig. 6. The identification of suboptimal cropland and the overlap with land degradation risk is a critical component of our cropland system optimization, and it is therefore introduced early in the abstract. Our analysis shows that approximately 66% of the identified suboptimal cropland is affected by various forms of land degradation risks (lines 131-132 and Fig. 3). This is an indication that agricultural activities are likely unsustainable, and the land requires interventions to contrast environmental degradation trends. In addition, other suboptimal cropland is fragmented and at steep slopes, which usually has environmental and economic challenges. Highlighting this information is instrumental to support targeted revegetation in suboptimal cropland. Further, “intensity reduction” was not intended as “reducing intensity on suboptimal croplands”, but rather to reduce agricultural intensity on *remaining non-suboptimal* cropland through cropland extensification, where patches of planted trees are integrated into high-density cropland. Here, cropland extensification could promote cascading benefits on yield through increased soil nutrient and water retention, reduced soil erosion, and improved climate resilience. Cropland extensification combined with crop switching to higher-yielding varieties can offset the reduction in crop production due to revegetation of suboptimal cropland. We have revised the abstract (lines 12-14) and main text (lines 77-80) to make this distinction clearer. We have now implemented a thorough revision of the abstract and of the introduction of the key terms used in the analysis, and we now believe the overall clarity of the paper has improved.

Regarding the last point of the comment, clarifications on the relationship between crop optimization, revegetation, and intensification. ‘Revegetation’ refers to natural regrowth or active afforestation on the suboptimal cropland (lines 73-74) and ‘crop optimization’ refers to strategic crop-switching optimization, prioritizing crops with high local suitability or dietary energy yield (lines 83-85). ‘Intensification’ refers to the conversion of low-density cropland (i.e., mixed parcels of crops and trees or shrub) to high-density cropland (i.e., pure cropland) (lines 77-80). The opposite term is ‘cropland extensification’, which involves integrating patches of planted trees into high-density cropland (lines 77-80). Their integration is based on a cropland system optimization that involves a spatially explicit combination of three components: (i) revegetating suboptimal croplands where factors like land degradation, steep terrain, or fragmentation make agricultural production unsustainable; (ii) implementing either intensification or extensification approaches on the rest of the cropland to compensate for the crop production losses resulting from such revegetation; and (iii) incorporating crop switching together with both intensification and extensification by selecting either the crop with the highest calorie supply or the one with the highest suitability, with the aim to compensate crop production losses from revegetation of suboptimal cropland. We now clarified all these

terms and their integration in the revised abstract, introduction (lines 70-85), and in the schematic representation of the research framework (Fig. 1). Further clarity comes from the addition of a new subsection of the Methods that is entirely dedicated to scenario description (lines 434-450).

3. For the body of paper, the scenario settings are not well explained, making the methodology difficult to follow. Due to these significant issues in clarity, terminology, and logical coherence, I do not recommend this paper for publication in its current form.

>> We have added a clearer description of the scenarios in a new subsection of the Methods (lines 434-450) and refer to Fig. 1 (and its caption) for the schematic representation of the research framework. The answer to the previous question also explains how the clarity and terminology have been improved.

4. In Figure 5, could you provide uncertainty estimates?

>> We thank the reviewer for highlighting the importance of uncertainty analysis. We have expanded the uncertainty analysis of our study and now include a more robust accounting of the different uncertainty sources and ranges. For carbon sequestration rates from natural regrowth and land clearing of above ground biomass in the intensification scenario, the original uncertainty layers provided by Cook-Patton et al. (2020) and Spawn et al. (2020), respectively, were used. Specifically, the uncertainty in natural regrowth data was derived from the standard deviation across 100 random forest model predictions per pixel, while for the above ground biomass, the uncertainty ranges are based on the cumulative standard error propagated from multiple data sources used in the modeling process. For the afforestation data lacking explicit uncertainty estimates, we assumed a default range of 15% of the pixel-scale estimated value, consistent with the uncertainty level of natural regrowth data ($\pm 13\%$ of the predicted value on average). Because the available datasets express their uncertainties in incompatible metrics, a uniform error-propagation scheme was not feasible. Instead, we constructed a conservative envelope to estimate the aggregated uncertainty of the datasets used, through pairing the lowest-bound estimate (mean - uncertainty) with the highest-bound estimate (mean + uncertainty), and vice versa. For visualization purposes, Fig. 5 shows the mean estimate, while the uncertainty ranges of the estimated impacts on carbon (and biodiversity) have been added to Fig. 6. A discussion of the uncertainty analysis performed has been added to the discussion section (lines 341-343), and a thorough explanation is given in the Supplementary text 3 dedicated to the uncertainties and limitations of the study.

Reference

Cook-Patton, S.C., Leavitt, S.M., Gibbs, D. *et al.* Mapping carbon accumulation potential from global natural forest regrowth. *Nature* **585**, 545–550 (2020). <https://doi.org/10.1038/s41586-020-2686-x>

Spawn, S.A., Sullivan, C.C., Lark, T.J. *et al.* Harmonized global maps of above and belowground biomass carbon density in the year 2010. *Sci Data* **7**, 112 (2020). <https://doi.org/10.1038/s41597-020-0444-4>

5. For all the simulated scenarios, how do you quantify the uncertainty range?

>> We further expanded our research and considered the main sources of uncertainty in the 1) identification of suboptimal cropland, 2) impact of crop yield fluctuations on the required area of cropland intensification or extensification, 3) quantification of the climate change mitigation benefits, and 4) biodiversity impact. These points are briefly discussed below and explained more in detail in the revised version of the manuscript.

1) Uncertainty in the identification of suboptimal cropland: We consider a variability in some key factors for the identification of suboptimal cropland and estimate an upper-bound and lower-bound area. For example, we considered various thresholds of slope gradients, patch sizes in identification of fragmented cropland, and productivity scores for suboptimal cropland. The parameters and corresponding changes in identified suboptimal cropland areas are shown in Tab. S5. As a result, the estimated area of suboptimal cropland ranged from 20.6 Mha (lower-bound estimation, Fig. S17) to 29.5 Mha (upper-bound estimation, Fig. S18), against 24.2 Mha reported in the main results. The revisions are presented in lines 335-338, 423-432, Supplementary text 3, Tab. S5, Fig. S17 and Fig. S18.

2) Impact of crop yield fluctuations on the required cropland area in the intensification and extensification scenarios: We calculated crop-specific yield fluctuations (expressed as absolute percentage changes) by considering the differences in yields for each crop across each of the 164 subregions between 2010 and present. These fluctuations at a subregional level are then used to infer the upper- and lower-bound yield variations for each grid within the corresponding subregion for the investigated crops. The resulting total crop production in suboptimal cropland ranges from 55.8 Pcal to 69.2 Pcal (61.3 Pcal reported in the main results), as shown in Fig. S19. Considering crop-specific yield fluctuations, the required areas and spatial patterns of cropland intensification or extensification under different scenarios change in magnitude, but their spatial patterns and characteristics remain broadly consistent with those described in the main text (lines 338-341, 605-

613), Supplementary text 3, and Fig. S19.

3) Uncertainty analysis in estimates of climate change mitigation potential. Please see our detailed response to Reviewer#1 comment 4. Briefly, we propagated the uncertainty across each source of carbon flow (carbon sequestration from trees in the revegetated suboptimal cropland and in the extensification scenarios, emissions from clearing land and afforestation activities, and fertilization-induced N₂O emissions), using the conservative envelope method. The resulting uncertainty ranges have been added to the main figure (Fig. 6).

4) Uncertainty analysis of cropland intensification or extensification's impact on biodiversity. The biodiversity impacts of the different scenarios are estimated as the difference in the local species richness between natural vegetation (used as benchmark) and other land cover types. Such estimation is based on spatial-explicit species richness data (Beyer et al., 2020, 2021) and meta-analysis comparing species richness difference across land-use types (Chaudhary et al., 2016; Torralba et al., 2016). The lower and upper bounds of the 95% confidence intervals reported in the meta-analysis are now used to estimate uncertainty ranges of biodiversity impacts (lines 736-738 for method). The resulting uncertainty bands are added to Fig. 6. Overall, our evaluation of biodiversity impact is broadly consistent with regional field observations. For instance, our analysis revealed that natural regrowth or afforestation on suboptimal cropland could enhance local species richness by $79.0 \pm 12.8\%$ and $24.8 \pm 8.81\%$, respectively. These findings align with previous site-based observations, which report that natural regrowth (or cropland abandonment) and afforestation could increase species richness of 51-78% and 18-41% on average, respectively (Tab. S9).

Overall, we find that the inclusion of this extended uncertainty analysis contributed to increasing the confidence and robustness of the results, providing an overview of the more sensitive factors and the possible variability in outcomes. Reflections on the matter have been added to the main text (discussion section, lines 330-350) and supplementary (Supplementary text 3: Uncertainties and limitations).

Reference

Beyer, R. & Manica, A. Global and country-level data of the biodiversity footprints of 175 crops and pasture. *Data Brief* 36 (2021). <https://doi.org:ARTN 106982>

Beyer, R. M. & Manica, A. Historical and projected future range sizes of the world's mammals, birds, and amphibians. *Nature Communications* 11 (2020).

Chaudhary, A., Burivalova, Z., Koh, L. et al. Impact of Forest Management on Species Richness: Global

Meta-Analysis and Economic Trade-Offs. *Sci Rep* 6, 23954 (2016). <https://doi.org/10.1038/srep23954>

Torralba, M., Fagerholm, N., Burgess, P. J., Moreno, G. & Plieninger, T. Do European agroforestry systems enhance biodiversity and ecosystem services? A meta-analysis. *Agr Ecosyst Environ* 230, 150-161 (2016).

6. Regarding equation 5, how do you calculate the N_c/b and NA_c/b ? Could you also provide a schematic figure for this part?

>> N_c and N_b represent the total carbon and biodiversity benefits, respectively, of scenarios relying solely on natural regrowth (N) in suboptimal cropland (i.e., I-HC-N, I-HS-N, E-HC-N and E-HS-N), while NA_c and NA_b represent the corresponding benefits of scenarios combining natural regrowth and afforestation (NA) (i.e., I-HC-NA, I-HS-NA, E-HC-NA and E-HS-NA). Both N_c and NA_c account for carbon sequestration from revegetated suboptimal cropland (natural regrowth only for N_c ; both natural regrowth and afforestation for NA_c) and trees established under extensification scenarios. They also include emissions from land clearing for intensification and afforestation activities, and N_2O emissions change induced by fertilization. Similarly, N_b and NA_b account for biodiversity pressure reduction from revegetated suboptimal cropland (natural regrowth only for N_b ; both natural regrowth and afforestation for NA_b), trees established under extensification scenarios, and increased biodiversity pressure from cropland intensification.

We have clarified the calculation procedure of this equation (lines 749-758) and a corresponding schematic figure has been created (Fig. S26).

Reviewer #2 (Remarks to the Author):

The paper is highly relevant and presents novel and significant findings at a continental (European) scale. It focuses on the identification and mapping of suboptimal cropland, with the goal of revegetating these areas to improve environmental sustainability without compromising food security. This conversion would contribute to improving biodiversity, carbon sequestration, and climate change, which are spatially estimated, through scenarios addressing both intensification and extensification or crop switching strategies to compensate for the loss of production resulting from the transformation of these suboptimal agricultural areas.

The work makes a valuable contribution to the field by taking a significant step toward identifying solutions and scenarios that can achieve food security without major environmental compromise. This approach would contribute to aligning with the objectives set out in the European Green Deal for 2030, without compromising food security or increasing land demand outside the EU. This vision is not only relevant but also urgent, given the current challenges of climate change and agricultural sustainability.

The central idea of the research is innovative and well-designed and executed. From a methodological standpoint, the approaches used, though I am not an expert in remote sensing, appear to be solid and well-founded. The methodology provides a robust basis for the replicability of the study. The results obtained clearly support the conclusions presented, suggesting that the study is consistent with its initial assumptions. Although there is an important section on the uncertainties and limitations of the study, it would still be valuable to consider a deeper discussion of the potential challenges and limitations of the models used. This would offer a more comprehensive and nuanced view of the practical and policy implications of the conclusions. For example, the authors have not considered the potential impacts of intensification on biodiversity in situ on cropland. However, the differences between wildlife-friendly farming and farming intensification are important when considering biodiversity, and this distinction should be more clearly addressed in the study.

In summary, the findings of this work are deserving of publication in a prestigious journal such as Nature Communications (NC), given their value and relevance both to the scientific community and to policy and environmental decision-making. The quality and potential impact of this study could make it a notable contribution to the field of agricultural sustainability and climate change.

>> Thank you for the general appreciation of our work and the useful feedback. In response, we have made several revisions to address the potential challenges and limitations of the models used, particularly regarding

biodiversity impacts associated with different cropland management strategies.

First, we considerably expanded the uncertainty analysis of our study, especially in quantitative terms. We quantified the uncertainties stemming from various parameters and datasets used in the analysis, such as: 1) suboptimal cropland area, considering various thresholds for its identification; 2) crop yield fluctuations and their influence on the area required for cropland intensification and extensification; 3) uncertainty in the estimates of climate change mitigation, by aggregating multiple uncertainty layers in the data used to model carbon flows, and 4) biodiversity impacts, by considering confidence intervals in the species richness difference across land-use types. All the details have been added to the discussion section of main text (lines 330-350), Supplementary text 3 dedicated to the uncertainties and limitations, and to additional figures in the supplementary file (Tab. S5, Fig. S14, S17, S18 and S19). The key figure of the analysis also contains uncertainty ranges (Fig. 6).

Second, we invested major efforts in comparing our results with existing literature to gain confidence in the outcomes of the analysis. We show the comparison with previous studies for potential estimates of European suboptimal cropland area (Tab. S6), carbon sequestration from natural regrowth (Tab. S7) and afforestation (Tab. S8), species richness impact after natural regrowth or afforestation in cropland (Tab. S9), and impact on crop yields of agricultural extensification (Tab. S1). In general, our findings are broadly in line with previous estimates reported for the same study area. The accuracy and validation of the primary datasets used are also detailed in the Methods section.

Regarding the distinction between wildlife-friendly farming and farming intensification, we have better clarified that our analysis largely takes them into account. Our extensification scenarios can also be interpreted as a form of wild-life friendly farming, as opposed to farming intensification. Biodiversity impacts are quantified using spatial-explicit species richness data (Beyer et al., 2020, 2021) and coefficients from a meta-analysis that synthesized results from 53 publications and 365 comparisons across five main biogeographical regions in Europe (Torralba et al., 2016). This meta-analysis shows that mixed landscapes of trees and cropland support 29.7% higher species richness (95% confidence interval: 18.7%–40.7%) compared to conventional cropland. This is now better explained in the paper (lines 717-721, and 736-738). Moreover, we used the lower-bound or upper-bound of the 95% confidence interval to conduct an uncertainty analysis (included in Fig. 6). We further benchmarked our evaluation of biodiversity impact with the existing literature in a dedicated table (Tab. S9), and we found that it is broadly consistent with regional field observations. For instance, our analysis revealed that natural regrowth or afforestation on suboptimal

cropland could enhance local species richness by $79.0 \pm 12.8\%$ and $24.8 \pm 8.81\%$, respectively. These findings align with previous observations, which report that natural regrowth (or cropland abandonment) and afforestation could increase 51-78% and 18-41% of local species richness, respectively.

References:

Beyer, R. & Manica, A. Global and country-level data of the biodiversity footprints of 175 crops and pasture. Data Brief 36 (2021). [https://doi.org:ARTN 106982](https://doi.org/ARTN 106982)

Beyer, R. M. & Manica, A. Historical and projected future range sizes of the world's mammals, birds, and amphibians. Nature Communications 11 (2020).

Torralba, M., Fagerholm, N., Burgess, P. J., Moreno, G. & Plieninger, T. Do European agroforestry systems enhance biodiversity and ecosystem services? A meta-analysis. Agr Ecosyst Environ 230, 150-161 (2016).

Reviewer #3 (Remarks to the Author):

1. This paper investigates how to reconcile agricultural production with climate action and biodiversity conservation in Europe by analyzing the potential for transforming suboptimal cropland. The authors identify 24.2 million hectares (Mha) of suboptimal cropland, which is underutilized and at risk of degradation, and evaluate different agricultural strategies (intensification and extensification) to preserve crop production while contributing to climate change mitigation and biodiversity conservation. The paper addresses a timely and critical issue regarding the balance between agricultural production, climate action, and biodiversity conservation through sustainable transformation. Overall, the science presented is sound, and the paper offers original insights, particularly in the context of the ongoing push to achieve the Sustainable Development Goals (SDGs) by 2030. I read the manuscript with great interest and appreciate the depth of analysis. However, some explanations, particularly in the discussion, seem oversimplified. I recommend accepting the manuscript after major revisions.

>> We are grateful for your insightful and encouraging comments, which helped us to strengthen the study. Despite the strict word limitations, we put efforts to provide more technical explanations and new insights into the discussion part, including 1) the impact mechanism of cropland extensification and intensification on crop yield, and climate change mitigation; 2) impact of crop-switching on nutrient supply (e.g., protein, macrominerals and micronutrients); 3) comparison with crop-switching strategy in existing literature; 4) policy feasibility and implication of our analysis; and 5) more quantitative uncertainty analysis.

Please see our detailed responses below.

General comments:

2. The paper addresses multiple issues with highly complex influencing factors. Agricultural profitability involves different crop varieties and management practices. This paper incorporates environmental impacts, including biodiversity conservation, greenhouse gas emissions, and land protection. The interconnections, quantification, and trade-offs among these factors contribute to a high degree of uncertainty in the results.

>> We have performed a more thorough quantitative uncertainty analysis and an in-depth comparison with existing studies. In particular, the quantitative uncertainty analysis considers possible variability in key parameters for 1) identification of suboptimal cropland, 2) impact of crop yield fluctuation on the area required in the cropland intensification and extensification scenarios, 3) estimates of climate change mitigation benefits and 4) biodiversity impact, and 5) a comparative analysis with previously published work.

Details are provided below.

1) Uncertainty in the identification of suboptimal cropland: We adjust threshold values for the identification of suboptimal cropland to estimate an upper-bound and lower-bound area. Variability is considered in key factors used in the estimate of cropland at steep slopes, fragmented, and with low productivity. Tab. S5 shows how much the identification of suboptimal cropland areas is sensitive to these factors, either independently or when considered simultaneously. The corresponding text and information in the revised manuscript are presented in the lines 335-338, 423-432, Supplement text 3, Tab. S5, and Fig S17 and S18. Briefly, the uncertainty analysis is implemented as follows:

- Slope cropland: A slope of 7° and 9° was tested as upper-bound and lower bound threshold, respectively, with 7° corresponding to the minimum identification standard of slope cropland from the different values used in previous similar research in Europe (as shown in Tab. S10).
- Fragmented cropland: Thresholds of either 20 or 10 ha in cropland area size are used to identify isolated cropland patches and achieve an upper-bound and lower-bound estimate, respectively. The threshold of approximately 10 ha is the highest resolution allowed for the detection of the individual patch size, as it corresponds to the grid-size resolution of the ESA CCI land cover dataset.
- Low-productivity cropland: It is identified by using an approach that measures vegetation productivity during the growing season with remotely-sensed data. Then, different z-scores are allocated across 164 subregions, based on the share of cropland relative to the total area within each subregion. In subregions with higher cropland shares (indicating more favorable cultivation conditions), lower z-scores were applied to determine the target proportion of low-productivity cropland. The values of z-score of -0.25 ~ -2 (normal distribution centered at -1.125), and of -0.75 ~ -2 (normal distribution centered at -1.375) were used as upper-bound estimation and lower-bound estimation.

As a result of explicitly considering uncertainty in these factors, the estimated area of suboptimal cropland ranges from 20.6 Mha (lower-bound estimation, Fig. S17) to 29.5 Mha (upper-bound estimation, Fig. S18), against 24.2 Mha reported in the main results. The spatial distribution patterns and total areas are broadly consistent with those presented in the main text (Fig. 2).

2) Impact of crop yield fluctuations on the required area of cropland intensification and extensification: We calculated crop-specific yield fluctuations (expressed as absolute percentage

changes) by considering the differences in yields for each crop across each of the 164 subregions between 2010 and present. These fluctuations at a subregional level are then used to infer the upper- and lower-bound yield variations for each grid within the corresponding subregion for the investigated crops. The resulting total crop production in suboptimal cropland ranges from 55.8 Pcal to 69.2 Pcal (61.3 Pcal reported in the main results). Considering crop-specific yield fluctuations, the required areas and spatial patterns of cropland intensification or extensification under different scenarios change in magnitude, but their spatial patterns and characteristics remain broadly consistent with those described in the main text (lines 338-341, 605-613, Supplement text 3, and Fig. S19).

- 3) Uncertainty analysis in estimates of climate change mitigation potentials.** We have added uncertainty ranges to the estimates of the climate change mitigation potentials of the 8 investigated scenarios (Fig. 6). Climate change mitigation benefits account for carbon sequestration by trees in the revegetated suboptimal cropland (via natural regrowth or afforestation) and in the cropland areas converted to tree cover in the extensification scenarios. Carbon emissions account for emissions from land clearing in the intensification scenarios (quantified by aboveground biomass carbon (AGBC) loss), life-cycle emissions from afforestation activities, and variations in soil N₂O emissions induced by changes in fertilizer inputs to suboptimal cropland and after crop-switching strategies. For carbon sequestration rates from natural regrowth and land clearing of above ground biomass in the intensification scenario, the original uncertainty layers provided by Cook-Patton et al. (2020) and Spawn et al. (2020), respectively, were used. Specifically, the uncertainty in natural regrowth data was derived from the standard deviation across 100 random forest model predictions per pixel, while for the above ground biomass the uncertainty ranges are based on the cumulative standard error propagated from multiple data sources used in the modeling process. For the afforestation data lacking explicit uncertainty estimates, we assumed a default range of 15% of the pixel-scale estimated value, consistent with the uncertainty level of natural regrowth data ($\pm 13\%$ of the predicted value on average). Because the available datasets express their uncertainties in incompatible metrics, a uniform error-propagation scheme was not feasible. Instead, we constructed a conservative envelope to estimate the aggregated uncertainty of the datasets used, through pairing the lowest-bound estimate (mean - uncertainty) with the highest-bound estimate (mean + uncertainty), and vice versa. The uncertainty ranges of the estimated impacts on carbon (and biodiversity) have been added to Fig. 6.

4) Uncertainty analysis of cropland intensification or extensification's impact on biodiversity:

The biodiversity impacts of the different scenarios are estimated as the difference in the local species richness between natural vegetation (used as benchmark) and other land cover types. Such estimation is based on spatial-explicit species richness data (Beyer et al., 2020, 2021) and meta-analysis comparing species richness differences across land-use types (Chaudhary et al., 2016; Torralba et al., 2016). The lower and upper bounds of the 95% confidence intervals reported in the meta-analysis are used as uncertainty ranges in our study to estimate biodiversity impacts (see lines 736-738 for method). The resulting uncertainty bands are added to Fig. 6. Overall, our evaluation of biodiversity impact is broadly consistent with regional field observations. For instance, our analysis revealed that natural regrowth or afforestation on suboptimal cropland could enhance local species richness by $79.0 \pm 12.8\%$ and $24.8 \pm 8.81\%$, respectively. These findings align with previous site-based observations, which report that natural regrowth (or cropland abandonment) and afforestation could increase species richness of 51-78% and 18-41% on average, respectively (Tab. S9).

5) Comparative analysis with previously published work

We invested major efforts in comparing our results with existing studies, so to validate the robustness of our analysis and explore possible divergences with previous literature. The comparison has been conducted for the individual key components of our analysis, such as estimates of European suboptimal cropland area (Tab. S6), carbon sequestration from natural regrowth (Tab. S7) and afforestation (Tab. S8), species richness difference between cropland and natural ecosystems (Tab. S9), and impact on crop yields of agricultural extensification (Tab. S1). In general, our estimate is broadly in line with previous findings reported for similar study areas.

We find that these additional investigations contribute to increasing the degree of confidence in the conclusions of our study, as the main results remain valid across a wide set of key uncertainty factors and in the benchmark with previous studies.

Reference

Beyer, R. & Manica, A. Global and country-level data of the biodiversity footprints of 175 crops and pasture. Data Brief 36 (2021). <https://doi.org:ARTN 106982>

Beyer, R. M. & Manica, A. Historical and projected future range sizes of the world's mammals, birds, and amphibians. Nature Communications 11 (2020).

Chaudhary, A., Burivalova, Z., Koh, L. et al. Impact of Forest Management on Species Richness: Global

Meta-Analysis and Economic Trade-Offs. *Sci Rep* 6, 23954 (2016). <https://doi.org/10.1038/srep23954>

Cook-Patton, S.C., Leavitt, S.M., Gibbs, D. et al. Mapping carbon accumulation potential from global natural forest regrowth. *Nature* 585, 545–550 (2020). <https://doi.org/10.1038/s41586-020-2686-x>

Spawn, S.A., Sullivan, C.C., Lark, T.J. et al. Harmonized global maps of above and belowground biomass carbon density in the year 2010. *Sci Data* 7, 112 (2020). <https://doi.org/10.1038/s41597-020-0444-4>

Torralba, M., Fagerholm, N., Burgess, P. J., Moreno, G. & Plieninger, T. Do European agroforestry systems enhance biodiversity and ecosystem services? A meta-analysis. *Agr Ecosyst Environ* 230, 150-161 (2016).

3. The paper comprehensively considers the interaction, or rather the compensatory relationship, between low-yield (suboptimal) farmland and high-yield farmland. For high-yield farmland, adaptive management for climate change needs to be implemented even in the absence of measures for low-yield farmland. Another possible scenario is that climate change mitigation subsidies could be provided directly to high-yield farmland managers, meaning there would be no interconnection between low-yield and high-yield farmland in this context.

>> Yes, we agree that adaptation to climate change is key to secure long-term high yields. However, our analysis is built around the treatment of the agricultural landscape as a coupled system, enabling compensatory adjustments between revegetation of suboptimal cropland and changes in other cropland to achieve optimal overall outcomes that try to preserve food production while enhancing climate change mitigation and biodiversity conservation. As such, the inclusion of a scenario based on adaptation of high-yield farming to future climatic conditions would fall outside this scope, requiring different approaches, tools and timescales for the assessment.

At the same time, we do find that climate change adaptive management in high-yield cropland should be pursued independently of low-yield (suboptimal) cropland interventions. We now explicitly acknowledge this issue as a limitation of our study, and we added some text to the discussion section to highlight the importance of developing adaptation plans to secure high yields in presence of climate change, leaving to future studies the tasks to design explorative scenarios for their implementation and quantification of environmental effects (lines 351-357).

4. The paper primarily focuses on the scientific value of farmland management adjustments. It is highly relevant to understanding agricultural activities and their associated environmental protection measures. However, is it feasible from a policy perspective? The paper examines Europe as a whole, but for policy implementation, feasibility would depend on financial subsidies. Are agricultural and environmental policies

independent in each European country? Does the EU framework provide a unified approach to environmental management and agricultural subsidies?

>> This is indeed a key point for the implementation of the most promising solutions. Generally, the landscape management strategies for cropland transition identified in our study – such as cropland extensification and revegetation of suboptimal cropland – are broadly consistent with the current direction of agricultural and environmental policies of Europe and individual member states. For example, the European Green Deal (EGD) and the Biodiversity Strategy for 2030 emphasize the restoration of high-diversity landscape features, such as hedgerows, buffer strips, and agroforestry plots, as integral components of climate adaptation and biodiversity conservation goals. A key economic instrument to support the sustainability transition in agriculture is the CAP (Common Agricultural Policy), which provides substantial funding to many EU countries. While national governments retain flexibility to design their own CAP Strategic Plans, they are required to contribute to the EU-wide objectives in biodiversity conservation, climate change mitigation, and sustainable land management. We tried to better elaborate on these aspects in the revised version of the manuscript. A paragraph has been added to the discussion section, where the policy feasibility underlying our analysis is discussed (lines 314-329). In the Supplementary Information, we document recent national cropland-related policies or strategies that have a key focus on revegetating suboptimal cropland and cropland extensification (Tab. S4). Together, they highlight a supportive policy environment across Europe for the scenarios investigated in this study. Our findings can stimulate actions and commitments in the public and decision makers.

Specific comments:

5. It would be beneficial to compare the findings with previous studies (e.g., Rising and Devineni, 2020; Xie et al., 2023), particularly in terms of how they model crop-switching strategies for sustainability. This comparison would help position the current work within the context of existing research and highlight its novel contributions.

>> We have enhanced comparison with findings from previous studies of crop-switching strategies (lines 293-303). However, a direct comparison is challenging owing to the different methods and datasets used across the studies, and there are some aspects to be considered when interpreting the various findings. Our investigated crop-switching strategy—favoring crops with the highest calorie supply or local suitability—emphasizes the cultivation of calorie-dense crops (e.g., maize and wheat) to reduce land competition to

facilitate achievement of climate change and biodiversity benefits. This approach is a compromise between maximizing cropland-use efficiency and respond to farmers' acceptance and local food preferences. Existing crop-switching studies often account for region-specific socioeconomic or environmental concerns, such as enhancing farm profitability in developing countries with high-return crops (Xie et al., 2023; Chakraborti et al., 2023; Wei et al., 2025), reducing water demand in water-scarce regions by replacing water-intensive crops with drought-tolerant species (Devineni et al., 2022; Chakraborti et al., 2023), or improving climate resilience in agricultural intensive areas (Risting et al., 2020; Gu et al., 2024). Some of these considerations have also been incorporated into our modeling framework (e.g., water constraints), but the current high-mechanization level of agriculture in Europe and the existence of supportive policies reduce the need to prioritize high-return crops. In addition, our analysis restricted crop-switching strategies in cropland intensification to regions without water-scarcity risk and with no high value for biodiversity. Our approach aims to enhance the environmental sustainability of cropland systems through leveraging landscape multifunctionality, providing a complementary solution to existing strategies that primarily focus on crop-specific traits.

Reference:

Chakraborti, R. et al. Crop switching for water sustainability in India's food bowl yields co-benefits for food security and farmers' profits. *Nat Water* 1 (2023).

Devineni, N., Perveen, S. & Lall, U. Solving groundwater depletion in India while achieving food security. *Nature Communications* 13 (2022).

Gu, W. Y. et al. Climate adaptation through crop migration requires a nexus perspective for environmental sustainability in the North China Plain. *Nature Food* 5 (2024).

Wei, D. Y., Castro, L. G., Chhatre, A., Tuninetti, M. & Davis, K. F. Swapping rice for alternative cereals can reduce climate-induced production losses and increase farmer incomes in India. *Nature Communications* 16 (2025).

Xie et al. Crop switching can enhance environmental sustainability and farmer incomes in China. *Nature* 616, (2023).

Rising, J. & Devineni, N. Crop switching reduces agricultural losses from climate change in the United States by half under RCP 8.5. *Nature Communications* 11 (2020).

6. While the manuscript focuses primarily on calorie production, it is crucial to also consider the nutritional aspects of crop transformation (Denning, 2025). Achieving universal food security requires not only sufficient calories but also access to nutritious food to address malnutrition and micronutrient deficiencies. The paper mainly discusses caloric outputs, but it would be valuable to explore how nutrition is integrated

into the crop-switching strategies. For example, Chakraborti et al. (2023) discuss how crop-switching can enhance environmental sustainability while improving nutrition and farmer livelihoods. I suggest adding a section that discusses how the proposed strategies can balance calorie production with the need for more nutritious crops.

>> Indeed, this can be an interesting aspect that can enrich the analysis. We have quantified the impacts of the assessed crop-switching strategies on the supply of protein, macrominerals (e.g., Sodium, Calcium, Magnesium, and Potassium), and micronutrients (e.g., Iron, Zinc, Copper, and Selenium). Please see lines 200-209 in the revised text for details. We make the corresponding data available in a new table of the supplementary material (Tab. S2).

Briefly, we find that the investigated crop-switching strategies increase the supply of certain macrominerals and micronutrients—including sodium, magnesium, zinc, and selenium—while causing minimal declines in protein and other nutrients. However, cereals, root and oil crops—the main crops involved in the switching—generally contribute less than 30% to daily intakes of protein, macrominerals, and micronutrients (Tab. S3), which are primarily sourced from animal-based foods and nutrient-dense crops (pulses, vegetables, fruits, and nuts). These nutrient-dense crops are retained in crop switching (e.g., their cultivation is preserved), meaning that the impact of the switching strategies on the supply of key nutrients is expected to be minimal. This is now clarified in the revised text (lines 200-209).

7. The manuscript shows the potential for CO₂ emissions reduction, but I recommend that the authors also consider other significant greenhouse gases, such as nitrous oxide (N₂O). N₂O, especially from fertilizer application, is a potent greenhouse gas with much higher global warming potential than CO₂ (Carlson et al., 2016; Tian et al., 2020). Not incorporating N₂O emissions could lead to an underestimation of the true climate impact of the proposed strategies, and it could impact the assessment of their overall effectiveness in climate mitigation.

>> Thank you for the input. We have now integrated changes in N₂O emissions into the calculation of climate change mitigation, through consideration of direct and indirect soil N₂O emissions from synthetic nitrogen fertilizer and animal manure applications, following the approach detailed in Carlson et al. (2017). These emissions were estimated using an updated version of the NL-N-RR model developed by Philibert et al. (2012) and Gerber et al. (2016), which was also adopted by Carlson et al. (2017). Estimations of indirect emissions were based on the 2006 IPCC Guidelines for National Greenhouse Gas Inventories (Table 11.3).

Briefly, results indicate that abandoning suboptimal cropland (excluding woody crops) could avoid fertilization-induced emissions of N₂O of 12.7 Mt CO₂-eq. yr⁻¹. In intensification scenarios, stopping growing the original crops in crop-switching regions saves 1.59–2.19 Mt CO₂-eq. yr⁻¹ of fertilization-induced emissions, while the cultivation of new crops in these regions generates 3.99–5.12 Mt CO₂-eq. yr⁻¹ of emissions (resulting in net emission increase of 2.41–2.93 Mt CO₂-eq. yr⁻¹). In extensification scenarios, there is a net decrease in soil N₂O emissions of 11.3–24.1 Mt CO₂-eq. yr⁻¹. These numbers are smaller than those connected to carbon fluxes but indeed are not negligible. They have now been incorporated into the analysis and embedded in Fig. 5 and Fig. 6. We have also updated the methods (lines 675-705) and results sections (lines 221-226, 235-239) accordingly, and added additional data to the supplementary information (Fig. S11-S14).

Reference:

Carlson, K. M. et al. Greenhouse gas emissions intensity of global croplands. *Nature Climate Change* 7, 63–68 (2017).

Gerber, J. S. et al. Spatially explicit estimates of N₂O emissions from croplands suggest climate mitigation opportunities from improved fertilizer management. *Global Change Biol* 22, 3383–3394 (2016).

IPCC 2006 IPCC Guidelines for National Greenhouse Gas Inventories. Prepared by the National Greenhouse Gas Inventories Programme (eds Eggleston, H. S., Buendia, L., Miwa, K., Ngara, T. & Tanabe, K.) Institute for Global Environmental Strategies, 2006.

Philibert, A., Loyce, C. & Makowski, D. Quantifying Uncertainties in N₂O Emission Due to N Fertilizer Application in Cultivated Areas. *Plos One* 7 (2012).

8. More discussion is needed on the mechanisms behind the results and how they align with past literature.

For instance, how do agricultural intensification and extensification balance the trade-off between food production and climate change mitigation potential? This would help clarify how the proposed strategies work in practice and their potential implications for broader climate and food security goals. I hope these comments contribute to improving the manuscript.

>> We have added a clearer explanation about the influence mechanisms of cropland extensification and intensification on crop production and climate change mitigation to the discussion (lines 270-292, 438-450).

Structurally complex landscapes that are a mix of trees and crops, as is the case for cropland extensification, help to moderate microclimatic conditions by buffering weather extremes, and enhances the populations of pollinators and natural pest predators, which in turn alleviate abiotic and biotic stress on crops' growth. The integration of trees also improves soil moisture and nutrient retention and enriches soil fertility through increased organic matter accumulation, thereby sustaining crop yields. Promotion of multifunctional

landscapes also support habitat diversity and multiple ecosystem services. There is spatial variability on these benefits, which also depend on the types of plant and crop species involved, but the general trends are consolidated in the scientific literature (Tab. S1 summarizes reported impact of agricultural extensification on crop yields). In the revised paper, the impact mechanisms of cropland extensification on crop yield and climate change mitigation are now better described in the discussion part mentioned above (line 270-292), as well as the Introduction (lines 37-42), and in the new section of the Methods dedicated to scenario description (lines 438-450). Regarding the connection with food production, the influence of the different scenarios on crop yields from the mechanisms mentioned above are estimated to assess how the total production can be kept constant while promoting climate change mitigation and biodiversity conservation on agricultural land. We find that the biodiversity conservation requirement and existing risk of water scarcity and land degradation restrict the available cropland for intensification (lines 168-171, and 273-278). Only relying on the cultivation of the locally dominant crops is insufficient to maintain crop production volumes constant through cropland intensification. Changing to either a crop type with higher calorie yields or expanding cultivation into these vulnerable areas is required.

9. References:

Carlson, K.M. et al., 2016. Greenhouse gas emissions intensity of global croplands. *Nature Climate Change*, 7(1): 63-68.

Chakraborti, R. et al., 2023. Crop switching for water sustainability in India's food bowl yields co-benefits for food security and farmers' profits. *Nature Water*, 1(10): 864-878.

Denning, G., 2025. Sustainable intensification of agriculture: the foundation for universal food security. *npj Sustainable Agriculture*, 3(1).

Rising, J. and Devineni, N., 2020. Crop switching reduces agricultural losses from climate change in the United States by half under RCP 8.5. *Nat Commun*, 11(1): 4991.

Tian, H. et al., 2020. A comprehensive quantification of global nitrous oxide sources and sinks. *Nature*, 586(7828): 248-256.

Xie, W. et al., 2023. Crop switching can enhance environmental sustainability and farmer incomes in China. *Nature*.

>> Thank you for providing this important literature, we have carefully reviewed it and included it in the revised manuscript where appropriate.

Reviewer #4 (Remarks to the Author):

>>We are grateful for sharing your expertise in the revision of manuscript and for the guidance provided.

Reviewer #5 (Remarks to the Author):

>>We are grateful for sharing your expertise in the revision of manuscript and for the guidance provided.

REVIEWER COMMENTS

Reviewer #1 (Remarks to the Author):

1. The authors have made a clear effort to address my concerns. The revised text is clear, and I am satisfied with the revisions.

>> Thank you for your encouraging words and insightful comments, which helped a lot in improving our manuscript.

Reviewer #2 (Remarks to the Author):

1. The authors have adequately responded to all the reviewers' concerns and revised the manuscript accordingly. Now the manuscript is much stronger. I find the current version satisfactory and have no additional comments.

>> Thank you for your encouraging words and insightful comments, which helped a lot in improving our manuscript.

Reviewer #3 (Remarks to the Author):

1. The authors have carefully revised the manuscript based on the reviewers' comments and suggestions. The current version is satisfactory and publishable. I would recommend acceptance for publication.

>> Thank you for your encouraging words and insightful comments, which helped a lot in improving our manuscript.

Reviewer #4 (Remarks to the Author):

>>We are grateful for sharing your expertise in the revision of manuscript and for the guidance provided.

Reviewer #5 (Remarks to the Author):

>>We are grateful for sharing your expertise in the revision of manuscript and for the guidance provided.